# Graph-based Discriminators: Sample Complexity and Expressiveness

**Roi Livni**
Tel Aviv University
rlivni@tauex.tau.ac.il

**Yishay Mansour**
Tel Aviv University and Google
mansour.yishay@gmail.com

## Abstract

A basic question in learning theory is to identify if two distributions are identical when we have access only to examples sampled from the distributions. This basic task is considered, for example, in the context of Generative Adversarial Networks (GANs), where a discriminator is trained to distinguish between a real-life distribution and a synthetic distribution. Classically, we use a hypothesis class $H$ and claim that the two distributions are distinct if for some $h \in H$ the expected value on the two distributions is (significantly) different.

Our starting point is the following fundamental problem: "is having the hypothesis dependent on more than a single random example beneficial". To address this challenge we define $k$-ary based discriminators, which have a family of Boolean $k$-ary functions $\mathcal{G}$. Each function $g \in \mathcal{G}$ naturally defines a hyper-graph, indicating whether a given hyper-edge exists. A function $g \in \mathcal{G}$ distinguishes between two distributions, if the expected value of $g$, on a $k$-tuple of i.i.d examples, on the two distributions is (significantly) different.

We study the expressiveness of families of $k$-ary functions, compared to the classical hypothesis class $H$, which is $k = 1$. We show a separation in expressiveness of $k + 1$-ary versus $k$-ary functions. This demonstrate the great benefit of having $k \geq 2$ as distinguishers.

For $k \geq 2$ we introduce a notion similar to the VC-dimension, and show that it controls the sample complexity. We proceed and provide upper and lower bounds as a function of our extended notion of VC-dimension.

## 1 Introduction

The task of discrimination consists of a *discriminator* that receives finite samples from two distributions, say $p_1$ and $p_2$, and needs to certify whether the two distributions are distinct. Discrimination has a central role within the framework of Generative Adversarial Networks [12], where a discriminator trains a neural net to distinguish between samples from a real-life distribution and samples generated synthetically by another neural network, called a *generator*.

A possible formal setup for discrimination identifies the discriminator with some distinguishing class $\mathcal{D} = \{f : X \to \mathbb{R}\}$ of *distinguishing functions*. In turn, the discriminator wishes to find the best $d \in \mathcal{D}$ that distinguishes between the two distributions. Formally, she wishes to find $d \in \mathcal{D}$ such that[1]

$$\left| \mathop{\mathbb{E}}_{x \sim p_1} [d(x)] - \mathop{\mathbb{E}}_{x \sim p_2} [d(x)] \right| > \sup_{d^* \in \mathcal{D}} \left| \mathop{\mathbb{E}}_{x \sim p_1} [d^*(x)] - \mathop{\mathbb{E}}_{x \sim p_2} [d^*(x)] \right| - \epsilon. \tag{1}$$

For examples, in GANs, the class of distinguishing functions we will consider could be the class of neural networks trained by the discriminator.

The first term in the RHS of eq. (1) is often referred to as the *Integral Probability Metric* (IPM distance) w.r.t a class $\mathcal{D}$ [16], denoted $\text{IPM}_\mathcal{D}$. As such, we can think of the discriminator as computing the $\text{IPM}_\mathcal{D}$ distance.

Whether two, given, distributions can be distinguished by the discriminator becomes, in the IPM setup, a property of the distinguishing class. Also, the number of examples needed to be observed will depend on the class in question. Thus, if we take a large expressive class of distinguishers, the discriminator can potentially distinguish between any two distributions that are far in total variation. In that extreme, though, the class of distinguishers would need to be very large and in turn, the number of samples needed to be observed scales accordingly. One could also choose a "small" class, but at a cost of smaller distinguishing power that yields smaller IPM distance.

For example, consider two distributions over $[n]$ to be distinguished. We could choose as a distinguishing class the class of *all* possible subsets over $n$. This distinguishing class give rise to the total variation distance, but the sample complexity turns out to be $O(n)$. Alternatively we can consider the class of *singletones*: This class will induce a simple IPM distance, with graceful sample complexity, however in worst case the IPM distance can be as small as $O(1/n)$ even though the total variation distance is large.

Thus, IPM framework initiates a study of generalization complexity where we wish to understand what is the expressive power of each class and what is its sample complexity.

For this special case that $\mathcal{D}$ consists of Boolean functions, the problem turns out to be closely related to the classical statistical learning setting and prediction [20]. The sample complexity (i.e., number of samples needed to be observed by the discriminator) is governed by a combinatorial measure termed *VC dimension*. Specifically, for the discriminator to be able to find a $d$ as in eq. (1), she needs to observe order of $\Theta(\frac{\rho}{\epsilon^2})$ examples, where $\rho$ is the VC dimension of the class $\mathcal{D}$ [5, 20].

In this work we consider a natural extension of this framework to more sophisticated discriminators: For example, consider a discriminator that observes pairs of points from the distribution and checks for collisions – such a distinguisher cannot apriori be modeled as a test of Boolean functions, as the tester measures a relation between two points and not a property of a single point. The collision test has indeed been used, in the context of synthetic data generation, to evaluate the *diversity* of the synthetic distribution [2].

More generally, suppose we have a class of 2-ary Boolean functions: $\mathcal{G} = \{g : g(x_1, x_2) \to \{0, 1\}\}$ and the discriminator wishes to (approximately) compute

$$\sup_{g \in \mathcal{G}} \left| \mathbb{E}_{(x_1, x_2) \sim p_1^2} [g(x_1, x_2)] - \mathbb{E}_{(x_1, x_2) \sim p_2^2} [g(x_1, x_2)] \right|. \tag{2}$$

Here $p^2$ denotes the product distribution over $p$. More generally, we may consider $k$-ary mappings, but for the sake of clarity, we will restrict our attention in this introduction to $k = 2$.

Such 2-ary Boolean mapping can be considered as graphs where $g(x_1, x_2) = 1$ symbolizes that there exists an edge between $x_1$ and $x_2$ and similarly $g(x_1, x_2) = 0$ denotes that there is no such edge. The collision test, for example, is modelled by a graph that contains only self–loops. We thus call such multi-ary statistical tests *graph-based distinguishers*.

Two natural question then arise

1. Do graph–based discriminators have any added distinguishing power over classical discriminators?

2. What is the sample complexity of graph–based discriminators?

With respect to the first question we give an affirmative answer and we show a separation between the distinguishing power of graph–based discriminators and classical discriminators. As to the second question, we introduce a new combinatorial measure (termed *graph VC dimension*) that governs the sample complexity of graph–based discriminators – analogously to the VC characterization of the sample complexity of classical discriminators. We next elaborate on each of these two results.

As to the distinguishing power of graph–based discriminators, we give an affirmative answer in the following sense: We show that there exists a single graph $g$ such that, for any distinguishing class $\mathcal{D}$ with bounded VC dimension, and $\epsilon$, there are two distributions $p_1$ and $p_2$ that are $\mathcal{D}$–indistinguishable but $g$ certifies that $p_1$ and $p_2$ are distinct. Namely, the quantity in eq. (2) is at least $1/4$ for $\mathcal{G} = \{g\}$.

This result may be surprising. It is indeed known that for any two distributions that are $\epsilon$–far in total variation, there exists a boolean mapping $d$ that distinguishes between the two distributions. In that sense, distinguishing classes are known to be universal. Thus, asymptotically, with enough samples any two distribution can be ultimately distinguished via a standard distinguishing function.

Nevertheless, our result shows that, given finite data, the restriction to classes with finite capacity is limiting, and there could be graph-based distinguishing functions whose distinguishing power is not comparable to *any* class with finite capacity. We stress that the same graph competes with *all* finite–capacity classes, irrespective of their VC dimension.

With respect to the second question, we introduce a new VC-like notion termed *graph VC dimension* that extends naturally to graphs (and hypergraphs). On a high level, we show that for a class of graph-based distinguishers with graph VC dimension $\rho$, $O(\rho)$ examples are sufficient for discrimination and that $\Omega(\sqrt{\rho})$ examples are necessary. This leaves a gap of factor $\sqrt{\rho}$ which we leave as an open question.

The notion we introduce is strictly weaker than the standard VC–dimension of families of multi-ary functions, and the proofs we provide do not follow directly from classical results on learnability of finite VC classes [20, 5]. In more details, a graph-based distinguishing class $\mathcal{G}$ is a family of Boolean functions over the product space of vertices $\mathcal{V}$: $\mathcal{G} \subseteq \{0,1\}^{\mathcal{V}^2}$. As such it is equipped with a VC dimension, the largest set of pairs of vertices that is shattered by $\mathcal{G}$.

It is not hard to show that finite VC is sufficient to achieve finite sample complexity bounds over 2-ary functions [9]. It turns out, though, that it is not a necessary condition: For example, one can show that the class of *k-regular graphs* has finite graph VC dimension but infinite VC dimension. Thus, even though they are not learnable in the standard PAC setting, they have finite sample complexity within the framework of discrimination.

The reason for this gap, between learnability and discriminability, is that learning requires uniform convergence with respect to any possible distribution over pairs, while discrimination requires uniform convergence only with respect to product distributions – formally then, it is a weaker task, and, potentially, can be performed even for classes with infinite VC dimension.

## 1.1 Related Work

The task of discrimination has been considered as early as the work of Vapnik and Chervonenkis in [20]. In fact, even though Vapnik and Chervonenkis original work is often referred in the context of prediction, the original work considered the question of when the empirical frequency of Boolean functions converges uniformly to the true probability over a class of functions. In that sense, this work can be considered as a natural extension to $k$-ary functions and generalization of the notion of VC dimension.

The work of [9, 8] studies also a generalization of VC theory to multi-ary functions in the context of ranking tasks and U-statistics. They study the standard notion of VC dimension. Specifically they consider the function class as Boolean functions over multi-tuples and the VC dimension is defined by the largest set of multi-tuples that can be shattered. Their work provides several interesting fast-rate convergence guarantees. As discussed in the introduction, our notion of capacity is weaker, and in general the results are incomparable.

**GANs**  A more recent interest in discrimination tasks is motivated by the framework of GANs, where a neural network is trained to distinguish between two sets of data – one is real and the other is generated by another neural network called *generator*. Multi-ary tests have been proposed to assess the quality of GANs networks. [2] suggests birthday paradox to evaluate *diversity* in GANs. [18] uses Binning to assess the solution proposed by GANs.

Closer to this work [15] suggests the use of a discriminator that observes samples from the $m$-th product distribution. Motivated by the problem of *mode collapse* they suggest a theoretical framework in which they study the algorithmic benefits of such discriminators and observe that they can

significantly reduce mode collapse. In contrast, our work is less concerned with the problem of mode collapse directly and we ask in general if we can boost the distinguishing power of discriminators via multi-ary discrimination. Moreover, we provide several novel sample complexity bounds.

**Property Testing**    A related problem to ours is that of testing closeness of distributions [3, 11]. Traditionally, testing closeness of distribution is concerned with evaluating if two discrete distributions are close vs. far/identical in *total variation*. [11], motivated by graph expansion test, propose a collision test to verify if a certain distribution is close to uniform. Interestingly, a collision test is a graph-based discriminator which turns out to be optimal for the setting[17]. Our sample–complexity lower bounds are derived from these results. Specifically we reduce discrimination to testing uniformity [17]. Other lower bounds in the literature can be similarly used to achieve alternative (yet incomparable bounds) (e.g. [7] provides a $\Omega(n^{2/3}/\epsilon^{3/4})$ lower bounds for testing whether two distributions are far or close).

In contrast with the aforementioned setup, here we do not measure distance between distributions in terms of total variation but in terms of an IPM distance induced by a class of distinguishers. The advantage of the IPM distance is that it sometimes can be estimated with limited amount of samples, while the total variation distance scales with the size of the support, which is often too large to allow estimation.

Several works do study the question of distinguishing between two distributions w.r.t a finite capacity class of tests, Specifically the work of [14] studies refutation algorithms that distinguish between noisy labels and labels that correlate with a bounded hypothesis class. [19] studies a closely related question in the context of realizable PAC learning. A graph-based discriminator can be directly turned to a refutation algorithm, and both works of [14, 19] show reductions from refutation to learning. In turn, the agnostic bounds of [14] can be harnessed to achieve lower bounds for graph-based discrimination. Unfortunately this approach leads to suboptimal lower bounds. It would be interesting to see if one can improve the guarantees for such reductions, and in turn exploit it for our setting.

## 2    Problem Setup

### 2.1    Basic Notations – Graphs and HyperGraphs

Recall that a $k$-hypergraph $g$ consists of a a set $\mathcal{V}_g$ of *vertices* and a collection of non empty $k$–tuples over $\mathcal{V}$: $E_g \subseteq \mathcal{V}^k$, which are referred to as *hyperedges*. If $k = 2$ then $g$ is called a graph. 1–hypergraphs are simply identified as subsets over $\mathcal{V}$. We will normally use $d$ to denote such 1-hypergraphs and will refer to them as *distinguishers*. A distinguisher $d$ can be identified with a Boolean function according to the rule: $d(x) = 1$ iff $x \in E_d$.

Similarly we can identify a $k$-hypergraph with a function $g : \mathcal{V}^k \to \{0, 1\}$. Namely, for any graph $g$ we identify it with the Boolean function

$$g(v_1, \ldots, v_k) = \begin{cases} 1 & (v_1, \ldots, v_k) \in E_g \\ 0 & \text{else} \end{cases}$$

We will further simplify and assume that $g$ is *undirected*, this means that for any permutation $\pi : [k] \to [k]$, we have that

$$g(v_{\pi(1)}, v_{\pi(2)}, \ldots, v_{\pi(k)}) = g(v_1, \ldots, v_k).$$

We will call undirected $k$-hypergraphs, $k$-distinguishers. A collection of $k$-distinguishers over a common set of vertices $\mathcal{V}$ will be referred to as a *k-distinguishing class*. If $k = 1$ we will simply call such a collection *a distinguishing class*. For $k > 1$ we will normally denote such a collection with $\mathcal{G}$ and for $k = 1$ we will often use the letter $\mathcal{D}$.

Next, given a distribution $P$ over vertices and a $k$–hypergraph $g$ let us denote as follows the frequency of an edge w.r.t $P$:

$$\mathbb{E}_P(g) = \underset{\mathbf{v}_{1:k} \sim P^k}{\mathbb{E}} [g(\mathbf{v}_{1:k})] = P^k \left[ \{ (\mathbf{v}_1, \ldots, \mathbf{v}_k) : (\mathbf{v}_1, \ldots, \mathbf{v}_k) \in E_g) \} \right],$$

where we use the notation $\mathbf{v}_{1:t}$ in shorthand for the sequence $(\mathbf{v}_1, \ldots, \mathbf{v}_t) \in \mathcal{V}^t$, and $P^k$ denotes the product distribution of $P$ $k$ times.

Similarly, given a sample $S = \{v_i\}_{i=1}^m$ we denote the empirical frequency of an edge:

$$\mathbb{E}_S(g) = \frac{1}{m^k} \sum_{\mathbf{u}_{1:k} \in S^k} g(\mathbf{u}_{1:k}) = \frac{|\{(\mathbf{u}_1, \ldots, \mathbf{u}_k) \in E_g : \forall i, \ \mathbf{u}_i \in S\}|}{m^k}$$

As a final set of notations: Given a $k$-hypergraph $g$ a sequence $\mathbf{v}_{1:n}$ where $n < k$, we define a $k - n$–distinguisher $g_{\mathbf{v}_{1:n}}$ as follows:

$$g_{\mathbf{v}_{1:n}}(\mathbf{u}_{1:k-n}) = g(\mathbf{v}_1, \ldots \mathbf{v}_n, \mathbf{u}_1, \ldots \mathbf{u}_{k-n}).$$

In turn, we define the following distinguishing classes: For every sequence $\mathbf{v}_{1:n}$, $n < k$, the distinguishing class $\mathcal{G}_{\mathbf{v}_{1:n}}$ is defined as follows:

$$\mathcal{G}_{\mathbf{v}_{1:n}} = \{g_{\mathbf{v}_{1:n}} : g \in \mathcal{G}\} \tag{3}$$

Finally, we point out that we will mainly be concerned with the case that $|\mathcal{V}| \leq \infty$ or $\mathcal{V} = \mathbb{N}$. However, all the results here can be easily extended to other domains as long as certain (natural) measurability assumptions are given to ensure that VC theory holds (see [20, 4]).

## 2.2 IPM distance

Given a class of distinguishers $\mathcal{D}$ the induced IPM distance [16], denoted by $\mathrm{IPM}_\mathcal{D}$, is a (pseudo)–metric between distributions over $\mathcal{V}$ defined as follows

$$\mathrm{IPM}_\mathcal{D}(p_1, p_2) = \sup_{d \in \mathcal{D}} |\mathbb{E}_{p_1}(d) - \mathbb{E}_{p_2}(d)| = \sup_{d \in \mathcal{D}} \left| \mathbb{E}_{v \sim p_1}[d(v)] - \mathbb{E}_{v \sim p_2}[d(v)] \right|.$$

The definition can naturally be extended to a general family of graphs, and we define:

$$\mathrm{IPM}_\mathcal{G}(p_1, p_2) = \sup_{g \in \mathcal{G}} |\mathbb{E}_{p_1}(g) - \mathbb{E}_{p_2}(g)]| = \sup_{g \in \mathcal{G}} \left| \mathbb{E}_{\mathbf{v}_{1:k} \sim p_1^k}[g(\mathbf{v}_{1:k})] - \mathbb{E}_{\mathbf{v}_{1:k} \sim p_2^k}[g(\mathbf{v}_{1:k})]] \right|$$

Another metric we would care about is the *total variation metric*. Given two distributions $p_1$ and $p_2$ the total variation distance is defined as:

$$\mathrm{TV}(p_1, p_2) = \sup_E |p_1(E) - p_2(E)|$$

where $E \subseteq \mathcal{V}^{\{0,1\}}$ goes over all measurable events.

In contrast with an IPM distance, the total variation metric is indeed a metric and any two distributions $p_1 \neq p_2$ we have that $\mathrm{TV}(p_1, p_2) > 0$. In fact, for every distinguishing class $\mathcal{D}$, $\mathrm{IPM}_\mathcal{D} \preceq \mathrm{TV}$.[2]

For finite classes of vertices $\mathcal{V}$, it is known that the total variation metric is given by

$$\mathrm{TV}(p_1, p_2) = \frac{1}{2} \sum_{v \in \mathcal{V}} |p_1(v) - p_2(v)|.$$

Further, if we let $\mathcal{D} = P(\mathcal{V})$ the power set of $\mathcal{V}$ we obtain

$$\mathrm{IPM}_{P(\mathcal{V})}(p_1, p_2) = \mathrm{TV}(p_1, p_2).$$

## 2.3 Discriminating Algorithms

**Definition 1.** *Given a distinguishing class $\mathcal{G}$ a $\mathcal{G}$-discriminating algorithm $A$ with sample complexity $m(\epsilon, \delta)$ is an algorithm that receives as input two finite samples $S = (S_1, S_2)$ of vertices and outputs a hyper-graph $g_S^A \in \mathcal{G}$ such that:*

*If $S_1, S_2$ are drawn IID from some unknown distributions $p_1, p_2$ respectively and $|S_1|, |S_2| > m(\epsilon, \delta)$ then w.p. $(1 - \delta)$ the algorithm's output satisfies:*

$$|\mathbb{E}_{p_1}(g_S^A) - \mathbb{E}_{p_2}(g_S^A)| > \mathrm{IPM}_{\mathcal{G}}(p_1, p_2) - \epsilon.$$

*The sample complexity of a class $\mathcal{G}$ is then given by the smallest possible sample complexity of a $\mathcal{G}$-discriminating algorithm $A$.*

*A class $\mathcal{G}$ is said to be discriminable if it has finite sample complexity. Namely there exists a discriminating algorithm for $\mathcal{G}$ with sample complexity $m(\epsilon, \delta) < \infty$.*

**VC classes are discriminable** For the case $k = 1$, discrimination is closely related to PAC learning. It is easy to see that a proper learning algorithm for a class $\mathcal{D}$ can be turned into a discriminator: Indeed, given access to samples from two distributions $p_1$ and $p_2$ we can provide a learner with labelled examples from a distribution $p$ defined as follows: $p(y = 1) = p(y = -1) = \frac{1}{2}$ and $p(\cdot|y = 1) = p_1$, and $p(\cdot|y = -1) = p_2$. Given access to samples from $p_1$ and $p_2$ we can clearly generate IID samples from the distribution $p$. If, in turn, we provide a learner with samples from $p$ and it outputs a hypothesis $d \in \mathcal{D}$ we have that (w.h.p):

$$
\begin{aligned}
|\mathbb{E}_{p_1}(d) - \mathbb{E}_{p_2}(d)| &= 2|\frac{1}{2} \underset{(x,y) \sim p_1 \times \{1\}}{\mathbb{E}} [yd(x)] + \frac{1}{2} \underset{(x,y) \sim p_2 \times \{-1\}}{\mathbb{E}} [yd(x)]| \\
&= 2| \underset{(x,y) \sim p}{\mathbb{E}} [yd(x)]| \\
&= 2(1 - 2p(d(x) \neq y)) \\
&\geq 2(1 - 2(\min_{d \in \mathcal{D}} p(d(x) \neq y) + \epsilon)) \\
&= \max_{d \in \mathcal{D}} (2(| \underset{(x,y) \sim p}{\mathbb{E}} yd(x)| - 4\epsilon) \\
&= \max_{d \in \mathcal{D}} |\mathbb{E}_{p_1}(d) - \mathbb{E}_{p_2}(d)| - 4\epsilon \\
&= \mathrm{IPM}_{\mathcal{D}}(p_1, p_2) - 4\epsilon
\end{aligned}
$$

One can also see that a converse relation holds, if we restrict our attention to learning balanced labels (i.e., $p(y = 1) = p(y = -1)$). Namely, given labelled examples from some balanced distribution, the output of a discriminator is a predictor that competes with the class of predictors induced by $\mathcal{D}$.

Overall, the above calculation, together with Vapnik and Chervonenkis's classical result [20] shows that classes with finite VC dimension $\rho$ are discriminable with sample complexity $O(\frac{\rho}{\epsilon^2})$.[3] The necessity of finite VC dimension for agnostic PAC-learning was shown in [1]. Basically the same argument shows that given a class $\mathcal{D}$, $\tilde{\Omega}(\frac{\rho}{\epsilon^2})$ examples are necessary for discrimination. We next introduce a natural extension of VC dimension to hypergraphs, which will play a similar role.

## 2.4 VC Dimension of hypergraphs

We next define the notion of graph VC dimension for hypergraphs, as we will later see this notion indeed characterizes the sample complexity of discriminating classes, and in that sense it is a natural extension of the notion of VC dimension for hypotheses classes:

**Definition 2.** *Given a family of $k$-hypergraphs, $\mathcal{G}$: The graph VC dimension of the class $\mathcal{G}$, denoted $\mathrm{gVC}(\mathcal{G})$, is defined inductively as follows: For $k = 1$ $\mathrm{gVC}(\mathcal{G})$ is the standard notion of VC dimension, i.e., $\mathrm{gVC}(\mathcal{G}) = \mathrm{VC}(\mathcal{G})$. For $k > 1$:*

$$\mathrm{gVC}(\mathcal{G}) = \max_{v \in \mathcal{V}} \{\mathrm{gVC}(\mathcal{G}_v)\}$$

Roughly, the graph VC dimension of a hypergraph is given by the VC dimension of the induced classes of distinguishers via projections. Namely, we can think of the VC dimension of hypergraphs as the projected VC dimension when we fix all coordinates in an edge except for one.

# 3 Main Results

We next describe the main results of this work. The results are divided into two sections: For the first part we characterize the sample complexity of graph–based distinguishing class. The second part is concerned with the expressive/distinguishing power of graph–based discriminators. All proofs are provided in the full version.

## 3.1 The sample complexity of graph-based distinguishing class

We begin by providing upper bounds to the sample complexity for discrimination

**Theorem 1** (Sample Complexity – Upper Bound). *Let $\mathcal{G}$ be a $k$–distinguishing class with $\mathrm{gVC}(\mathcal{G}) = \rho$ then $\mathcal{G}$ has sample complexity $O(\frac{\rho k^2}{\epsilon^2} \log 1/\delta)$.*

Theorem 1 is a corollary of the following uniform convergence upper bound for graph-based distinguishing classes.

**Theorem 2** (uniform convergence). *Let $\mathcal{G}$ be a $k$–distinguishing class with $\mathrm{gVC}(\mathcal{G}) = \rho$. Let $S = \{v_i\}_{i=1}^m$ be an IID sample of vertices drawn from some unknown distribution $P$. If $m = \Omega(\frac{\rho k^2}{\epsilon^2} \log 1/\delta)$ then with probability at least $(1 - \delta)$ (over the randomness of S):*

$$\sup_{g \in \mathcal{G}} |\mathbb{E}_S(g) - \mathbb{E}_P(g)| \leq \epsilon.$$

We next provide a lower bound for the sample complexity of discriminating algorithms in terms of the graph VC dimension of the class

**Theorem 3** (Sample Complexity – Lower Bound). *Let $\mathcal{G}$ be a $k$–distinguishing class with $\mathrm{gVC}(\mathcal{G}) = \rho$. Any $\mathcal{G}$-discriminating algorithm with accuracy $\epsilon > 0$ that succeeds with probability $1 - \frac{2^{-k \log k}}{3}$, must observe at least $\Omega\left(\frac{\sqrt{\rho}}{2^{7k^3}\epsilon^2}\right)$ samples.*

Our upper bounds and lower bounds leave a gap of order $O(\sqrt{\rho})$: As dicussed in section 2.3, for the case $k = 1$ we can provide a tight $\theta(\frac{\rho}{\epsilon^2})$ bound through a reduction to agnostic PAC learning and the appropriate lower bounds[1].

## 3.2 The expressive power of graph-based distinguishing class

So far we have characterized the discriminability of graph-based distinguishing classes. It is natural though to ask if graph–based distinguishing classes add any advantage over standard 1-distinguishing classes. In this section we provide several results that show that indeed graph provide extra expressive power over standard distinguishing classes.

We begin by providing a result over infinite graphs.

**Theorem 4.** *Let $\mathcal{V} = \mathbb{N}$. There exists a distinguishing graph class $\mathcal{G}$, with sample complexity $m(\epsilon, \delta) = O(\frac{\log 1/\delta}{\epsilon^2})$ (in fact $|\mathcal{G}| = 1$) such that: for any 1-distinguishing class $\mathcal{D}$ with finite VC dimension, and every $\epsilon > 0$ there are two distributions $p_1, p_2$ such that $\mathrm{IPM}_{\mathcal{D}}(p_1, p_2) < \epsilon$ but $\mathrm{IPM}_{\mathcal{G}}(p_1, p_2) > 1/2$*

Theorem 4 can be generalized to higher order distinguishing classes :

**Theorem 5.** *Let $\mathcal{V} = \mathbb{N}$. There exists a $k$-distinguishing class $\mathcal{G}_k$, with sample complexity $m(\epsilon, \delta) = O(\frac{k^2 + \log 1/\delta}{\epsilon^2})$ such that: For any $k-1$-distinguishing class $\mathcal{G}_{k-1}$ with bounded sample complexity, and every $\epsilon > 0$ there are two distributions $p_1, p_2$ such that $\mathrm{IPM}_{\mathcal{G}_{k-1}}(p_1, p_2) < \epsilon$ and $\mathrm{IPM}_{\mathcal{G}_k}(p_1, p_2) > 1/4$.*

**Finite Graphs**   We next study the expressive power of distinguishing graphs over finite domains.

It is known that, over a finite domain $\mathcal{V} = \{1, \ldots, n\}$, we can learn with a sample complexity of $O(\frac{n}{\epsilon^2} \log 1/\delta)$ any distinguishing class. In fact, we can learn the total variation metric (indeed the sample complexity of $\mathcal{P}(\mathcal{V})$ is bounded by $\log |\mathcal{P}(\mathcal{V})| = n$).

Therefore if we allow classes whose sample complexity scales linearly with $n$ we cannot hope to show any advantage for distinguishing graphs. However, in most natural problems $n$ is considered to be very large (for example, over the Boolean cube $n$ is exponential in the dimension). We thus, in general, would like to study classes that have better complexity in terms of $n$. In that sense, we can show that indeed distinguishing graphs yield extra expressive power.

In particular, we show that for classes with sublogarithmic sample complexity, we can construct graphs that are incomparable with a higher order distinguishing class.

**Theorem 6.** *Let $|\mathcal{V}| = n$. There exists a $k$-distinguishing class $\mathcal{G}_k$, with sample complexity $m(\epsilon, \delta) = O(\frac{k^2 + \log 1/\delta}{\epsilon^2})$ (in fact $|\mathcal{G}| = 1$) such that: For any $\epsilon > 0$ and any $k - 1$ distinguishing class $\mathcal{G}_{k-1}$ if:*

$$\mathrm{IPM}_{\mathcal{G}_{k-1}} \succ \epsilon \cdot \mathrm{IPM}\mathcal{G}_k$$

*then $\mathrm{gVC}(\mathcal{G}_{k-1}) = \Omega(\frac{\epsilon^2}{k^2}\sqrt{\log n})$.*

We can improve the bound in theorem 6 for the case $k = 1$ .

**Theorem 7.** *Let $|\mathcal{V}| = n$. There exists a $2$-distinguishing class $\mathcal{G}$, with sample complexity $m(\epsilon, \delta) = O(\frac{\log 1/\delta}{\epsilon^2})$ (in fact $|\mathcal{G}| = 1$) such that: For any $\epsilon > 0$ and any distinguishing class $\mathcal{D}$ if:*

$$\mathrm{IPM}_{\mathcal{D}} \succ \epsilon \cdot \mathrm{IPM}\mathcal{G}$$

*then $\mathrm{gVC}(\mathcal{D}) = \tilde{\Omega}(\epsilon^2 \log n)$.*

## 4 Discussion and open problems

In this work we developed a generalization of the standard framework of discrimination to graph-based distinguishers that discriminate between two distributions by considering multi-ary tests. Several open question arise from our results:

**Improving Sample Complexity Bounds**    In terms of sample complexity, while we give a natural upper bound of $O(\rho k^2)$, the lower bound we provide are not tight neither in $d$ nor in $k$ and we provide a lower bound of $\Omega(\frac{\sqrt{\rho}}{2^{poly(k)}})$ This leave room for improvement both in terms of $\rho$ and in terms of $k$.

**Improving Expressiveness Bounds**    We also showed that, over finite domains, we can construct a graph that is incomparable with any class with VC dimension $\Omega(\epsilon^2 \log n)$. The best upper bound we can provide (the VC of a class that competes with any graph) is the naive $O(n)$ which is the VC dimension of the total variation metric.

Additionally, for the $k$-hypergraph case, our bounds deteriorate to $\Omega(\epsilon^2 \sqrt{\log n})$. The improvement in the graph case follows from using an argument in the spirit of Boosting [10] and Hardcore Lemma [13] to construct two indistinguishable probabilities with distinct support over a small domain. It would be interesting to extend these techniques in order to achieve similar bounds for the $k > 2$ case.

**Relation to GANs and Extension to Online Setting**    Finally, a central motivation for learning the sample complexity of discriminators is in the context of GANs. It then raises interesting questions as to the *foolability* of graph-based distinguishers.

The work of [6] suggests a framework for studying sequential games between generators and discriminators (*GAM-Fooling*). In a nutshell, the GAM setting considers a sequential game between a generator $G$ that outputs distributions and a discriminator $D$ that has access to data from some distribution $p^*$ (not known to $G$). At each round of the game, the generator proposes a distribution and the discriminator outputs a $d \in \mathcal{D}$ which distinguishes between the distribution of $G$ and the true distribution $p^*$. The class $\mathcal{D}$ is said to be GAM-Foolable if the generator outputs after finitely many rounds a distribution $p$ that is $\mathcal{D}$–indistinguishable from $p^*$

[6] showed that a class $\mathcal{D}$ is GAM–foolable if and only if it has finite Littlestone dimension. We then ask, similarly, which classes of graph–based distinguishers are GAM-Foolable? A characterization of such classes can potentially lead to a natural extension of the Littlestone notion and online prediction, to graph-based classes analogously to this work w.r.t VC dimension

**Acknoweledgements**    Y.M is supported in part by a grant from the ISF

## Footnotes

[1]Note that with such $d$ at hand, with an order of $O(1/\epsilon^2)$ examples one can verify if any discriminator in the class certifies that the two distributions are distinct.

[2]we use the notation $f_1 \preceq f_2$ to denote that for every $x, y$ we have $f_1(x, y) \leq f_2(x, y)$.

[3]Recall that the VC dimension of a class $\mathcal{D}$ is the largest set that can be shattered by $\mathcal{D}$ where a set $S \subseteq \mathcal{V}$ is said to be shattered if $\mathcal{D}$ restricted to $S$ consists of $2^{|S|}$ possible Boolean functions.

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
