[Supplementary Material · full.pdf]

# Graph-based Discriminators: Sample Complexity and Expressiveness

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

# A  Preliminineries and Technical Background

## A.1  Statistical Learning Theory

We begin with a brief overview of some classical results in Statistical Learning theory which characterizes VC classes. Throughout we assume a domain $\mathcal{X}$ and a *hypothesis class* which is a family of Boolean functions over $\mathcal{X}$: $\mathcal{H} \subseteq \{0,1\}^{\mathcal{X}}$.

**Theorem 8.** *[Within proof of Thm. 6.11 in [20]] Let $\mathcal{H}$ be a class with VC dimension $\rho$ then*

$$\mathop{\mathbb{E}}_{S \sim D^m} \left[ \sup_{h \in \mathcal{H}} |\mathbb{E}_S(h) - \mathbb{E}_D(h)| \right] \leq \frac{4 + \sqrt{\rho \log(2em/\rho)}}{\sqrt{2m}}$$

Recall that a class $\mathcal{H}$ has the *uniform convergence property*, if for some $m : (0,1)^2 \to \mathbb{N}$ if $P$ is some unknown distribution and $S = \{x_i\}_{i=1}^m$ is a sample drawn IID from $P$ such that $|S| > m(\epsilon, \delta)$ then w.p. $(1 - \delta)$ (over the sample $S$):

$$|\frac{1}{m} \sum_{i=1}^m h(x_i) - \mathop{\mathbb{E}}_{x \sim P}[h(x)]| < \epsilon$$

The following, high probability analogue of theorem 8, is also an immediate corollary of Theorem 6.8 in [20][4]:

**Corollary 1.** *[Within Thm 6.8 [20]] Let $\mathcal{D}$ be a class with VC dimension $\rho$. There exists a constant $C > 0$, such that:*

*Let $p$ be a distribution with finite support over $\mathcal{V}$. Let $S$ be an IID sequence of $m$ elements drawn from p, and denote by $p_S$ the empirical distribution over S. If $m \geq C\frac{\rho + \log 1/\delta}{\epsilon^2}$ then w.p. $(1 - \delta)$ (over the random choice of S) we have that*

$$\mathrm{IPM}_{\mathcal{D}}(p, p_S) = \sup_{d \in \mathcal{D}} |\mathbb{E}_p(d) - \mathbb{E}_{p_S}(d)| < \epsilon$$

## A.2  Closeness Testing for Discrete Distribution

The problem of testing the closeness of two discrete distributions can be phrased as follows: Given samples from two distributions $p_1$ and $p_2$ the tester needs to distinguish between the case $p_1 = p_2$ and the case that $\|p_1 - p_2\|_1 \geq \epsilon$. We will rely on the following result due to [18] (see also [7] for discussion).

**Theorem 9.** *Given $\epsilon > 0$ and access to samples from distributions $p_1$ and $p_2$ over $[n]$ any algorithm that returns with probability $2/3$ $EQUIVALENT$ if $p_1 = p_2$ and returns $DISTINCT$ if $\|p_1 - p_2\|_1 > \epsilon$ must observe at least $\Omega\left(\sqrt{n}/\epsilon^2\}\right)$ samples.*

We note that [7] gives a slightly better lower bound, of $\Omega\left(\max(n^{3/4}/\epsilon^{4/3}, \sqrt{n}/\epsilon^2)\right)$. However, our proofs exploit other processes that exploit concentration inequalities, and it will be simpler to focus on rates of order $O(1/\epsilon^2)$.

 # B  Sample Complexity –Proofs

 ## B.1  Proof of theorem 2

 **Theorem 2** (uniform convergence). *Let $\mathcal{G}$ be a $k$–distinguishing class with $\mathrm{gVC}(\mathcal{G}) = \rho$. Let*
 *$S = \{v_i\}_{i=1}^m$ be an IID sample of vertices drawn from some unknown distribution $P$. If $m =$*
 *$\Omega(\frac{\rho k^2}{\epsilon^2} \log 1/\delta)$ then with probability at least $(1-\delta)$ (over the randomness of $S$):*

$$\sup_{g \in \mathcal{G}} |\mathbb{E}_S(g) - \mathbb{E}_P(g)| \leq \epsilon$$

 Fix a $k$–distinguishing class $\mathcal{G}$ with graph VC dimension $\rho$. As in the standard proof of uniform
 convergence for VC classes, we first prove the statement in expectation and then apply Mcdiarmid's
 inequality to prove the result w.h.p. Specifically, we will use the following Lemma (whose proof is
 given in appendix B.1.1):

 **Lemma 1** (Uniform Convergence in Expectation). *Let $\mathcal{G}$ be a $k$–distinguishing class with $\mathrm{gVC}(\mathcal{G}) =$*
 *$\rho$. Let $S = \{v_i\}_{i=1}^m$ be an IID sample of vertices drawn from some unknown distribution $P$. Then,*

$$\mathbb{E}_{S \sim P^m} \left[ \sup_{g \in \mathcal{G}} |\mathbb{E}_S(g) - \mathbb{E}_D(g)| \right] \leq \frac{k\sqrt{4 + \rho \log(2em/\rho)}}{\sqrt{2m}} + \frac{k(k-1)}{m}$$

 We next proceed with the proof of theorem 2, assuming the correctness of lemma 1. Define

$$F(S) = \sup_{g \in \mathcal{G}} |\mathbb{E}_S(g) - \mathbb{E}_D(g)|,$$

 Let $S = (v_1, \ldots, v_m)$ be a sample and $S'$, some sequence that differ from $S$ only in the $i$-th vertex
 then we will show that:

$$|F(S) - F(S')| \leq \frac{2k}{m} \tag{4}$$

 Once we show eq. (4) holds, the result indeed follow from Mcdiarmid's inequality and lemma 1.
 Specifically if we assume that $m \geq \frac{8k^2(4+\rho\log(2em/\rho))}{\epsilon^2} + \frac{2k^2 1/\delta}{\epsilon^2}$ then we obtain from lemma 1 that
 in expectation:

$$\mathbb{E}_{S \sim D^m} \sup_{g \in \mathcal{G}} |\mathbb{E}_S(g) - \mathbb{E}_D(g)| \leq \frac{\epsilon}{2}$$

 Applying Mcdiarmid's we obtain that with probability at least $(1 - e^{-\frac{m\epsilon^2}{8k^2}})$, over the sample $S$:

$$F(S) - \mathbb{E}_S[F(S)] = \sup_{g \in \mathcal{G}} |\mathbb{E}_S(g) - \mathbb{E}_D(g)| - \mathbb{E}_{S \sim D^m} \sup_{g \in \mathcal{G}} |\mathbb{E}_S(g) - \mathbb{E}_D(g)| \leq \frac{\epsilon}{2}.$$

 Noting that $m > \frac{8k^2 \log 1/\delta}{\epsilon^2}$, we obtain that with probability at least $(1-\delta)$

$$F(S) = \sup_{g \in \mathcal{G}} |\mathbb{E}_S(g) - \mathbb{E}_D(g)| \leq \mathbb{E}_{S \sim D^m} \sup_{g \in \mathcal{G}} |\mathbb{E}_S(g) - \mathbb{E}_D(g)| + \frac{\epsilon}{2} \leq \epsilon$$

 We are thus left with proving that eq. (4) holds.

 For an index $i$ and $m \geq i$, let us denote by $\pi_{i,m}$ all $k$-subsets of indices from $\{1, \ldots, m\}$ that include
 $i$ and we let $\pi_{\neg i,m}$ be all $k$-sequences that do not include $i$. Given a set $S$ of size $m$ let $S_{i,+}$ all the
 $k$-subsets of $S$ that include $v_i$ and let $S_{i,-}$ be all the $k$-subsets that do not include $v_i$. Next, denote

$$L_{S_{i,+}}(g) = \frac{1}{m^k} \sum_{(i_1,\ldots,i_k) \in \pi_{i,m}} g(\mathbf{u}_{i_1}, \ldots, \mathbf{u}_{i_k})$$

 And similarly

$$L_{S_{i,-}}(g) = \frac{1}{m^k} \sum_{(i_1,\ldots,i_k) \in \pi_{\neg i,m}} g(\mathbf{u}_{i_1}, \ldots, \mathbf{u}_{i_k})$$

 Then, let $S$ and $S'$ be two samples that differ on the $i$-th example. Specifically assume that $v_i \in S$
 and $v_i' \in S'$. Note that $S_{i,-} = S'_{i,-}$. Then:

$$F(S) - F(S') = \sup_{g \in \mathcal{G}} |\mathbb{E}_S(g) - \mathbb{E}_D(g)| - \sup_{g \in \mathcal{G}} |\mathbb{E}_{S'}(g) - \mathbb{E}_D(g)|$$

$$= \sup_{g \in \mathcal{G}} |L_{S_{i,+}}(g) + L_{S_{i,-}}(g) - \mathbb{E}_D(g)| - \sup_{g \in \mathcal{G}} |L_{S'_{i,+}}(g) + L_{S'_{i,-}}(g) - \mathbb{E}_D(g)|$$

$$\leq \sup_{g \in \mathcal{G}} |L_{S_{i,+}}(g) + L_{S_{i,-}}(g) - \mathbb{E}_D(g) - (L_{S'_{i,+}}(g) + L_{S'_{i,-}}(g) - \mathbb{E}_D(g))|$$

$$= \sup_{g \in \mathcal{G}} |L_{S_{i,+}}(g) - L_{S'_{i,+}}(g)|$$

$$= \sup_{g \in \mathcal{G}} |L_{S_{i,+}}(g)| + |\sup_{g \in \mathcal{G}} |L_{S'_{i,+}}(g)|$$

$$\leq \frac{|S_{i,+}|}{m^k} + \frac{|S'_{i,+}|}{m^k}$$

$$= 2\frac{m^k - (m-1)^k}{m^k}$$

$$= 2 - 2(1 - \frac{1}{m})^k$$

$$\leq 2\frac{k}{m}$$

We are thus left with proving lemma 1:

### B.1.1 Proof of lemma 1

The proof of the statement follows by induction. The case $k = 1$ is the standard uniform convergence property of VC classes, and it follows from theorem 8.

We next proceed to prove the statement for $k$, assuming it holds for $k - 1$. We begin by exploiting trinagular inequality and together with adding/substracting terms:

$$\mathbb{E}_{S \sim D^m} \left[ \sup_{g \in \mathcal{G}} |\mathbb{E}_S(g) - \mathbb{E}_D(g)| \right]$$

$$= \mathbb{E}_{S \sim D^m} \left[ \sup_{g \in \mathcal{G}} |\mathbb{E}_S(g) - \frac{1}{m^{k-1}} \sum_{\mathbf{v}_{1:k-1} \in S^{k-1}} \mathbb{E}_v g_{\mathbf{v}_{1:k-1}}(v) + \frac{1}{m^{k-1}} \sum_{\mathbf{v}_{1:k-1} \in S^{k-1}} \mathbb{E}_v g_{\mathbf{v}_{1:k-1}}(v) - \mathbb{E}_D(g)| \right]$$

$$\leq \underbrace{\mathbb{E}_{S \sim D^m} \left[ \sup_{g \in \mathcal{G}} |\mathbb{E}_S(g) - \frac{1}{m^{k-1}} \sum_{\mathbf{v}_{1:k-1} \in S^{k-1}} \mathbb{E}_v g_{\mathbf{v}_{1:k-1}}(v)| \right]}_{*}$$

$$+$$

$$\underbrace{\mathbb{E}_{S \sim D^m} \left[ \sup_{g \in \mathcal{G}} |\frac{1}{m^{k-1}} \sum_{\mathbf{v}_{1:k-1} \in S^{k-1}} \mathbb{E}_v g_{\mathbf{v}_{1:k-1}}(v) - \mathbb{E}_D(g)| \right]}_{**}$$

We next bound the two terms

**Bounding \***

$$\mathbb{E}_{S\sim D^m}\left[\sup_{g\in\mathcal{G}}\left|\frac{1}{m^{k-1}}\sum_{\mathbf{v}_{1:k-1}\in S^{k-1}}\frac{1}{m}\sum_{v\in S}g_{\mathbf{v}_{1:k-1}}(v)-\frac{1}{m^{k-1}}\sum_{\mathbf{v}_{1:k-1}\in S^{k-1}}\mathbb{E}_v\,g_{\mathbf{v}_{1:k-1}}(v))\right|\right]$$

$$\leq\mathbb{E}_{S\sim D^m}\left[\frac{1}{m^{k-1}}\sum_{\mathbf{v}_{1:k-1}\in S^{k-1}}\sup_{d\in\mathcal{G}_{\mathbf{v}_{1:k-1}}}\left|\frac{1}{m}\sum_{v\in S}d(v)-\mathbb{E}_v\,d(v)\right|\right]$$

$$=\mathbb{E}_{S\sim D^m}\left[\mathbb{E}_{\mathbf{v}_{1:k}\sim\mathcal{U}_{S^{k-1}}}\sup_{d\in\mathcal{G}_{\mathbf{v}_{1:k-1}}}\left|\frac{1}{m}\sum_{v\in S}d(v)-\mathbb{E}_v\,d(v)\right|\right]$$

429 where we denoted by $\mathcal{U}_{S^{k-1}}$ the uniform distribution over $k$-tuples from $S$. The expectation in the last
430 expression is thus taken w.r.t a process where we pick $m$ elements according to $d$ and then partition
431 them to $m-k+1$ elements and to a sequence $\mathbf{v}_{1:k-1}$ of distinct element. This process is equivalent
432 to simply choosing $m-k+1$ elements according to $D$, and then picking $k-1$ new elements, again,
433 according to $D$. We thus continue and write:

$$=\mathbb{E}_{S\sim D^{m-k+1}}\mathbb{E}_{(\mathbf{v}_1,\dots,\mathbf{v}_{k-1})\sim D^{k-1}}\left[\sup_{d\in\mathcal{G}_{\mathbf{v}_{1:k-1}}}\left|\frac{1}{m}\sum_{v\in S}d(v)+\frac{1}{m}\sum_{i=1}^{k-1}d(\mathbf{v}_i)-\mathbb{E}_v\,d(v)\right|\right]$$

$$=\mathbb{E}_{S\sim D^{m-k+1}}\mathbb{E}_{(\mathbf{v}_1,\dots,\mathbf{v}_{k-1})\sim D^{k-1}}\left[\sup_{d\in\mathcal{G}_{\mathbf{v}_{1:k-1}}}\left|\frac{1}{m}\sum_{v\in S}d(v)-\mathbb{E}_v\,d(v)+\frac{1}{m}\sum_{i=1}^{k-1}d(\mathbf{v}_i)\right|\right]$$

434 Note that the quantity $\frac{1}{m}\sum d(\mathbf{v}_i)$ is dependent on $\mathcal{G}_{\mathbf{v}_{1:k-1}}$, namely these are random sampled choices
435 that depend on our choice of distinguishing class. To bound their effect we next add and subtract
436 auxiliary random variables $\mathbf{u}_1,\dots,\mathbf{u}_{k-1}$ sampled IID according to $D$:

$$=\mathbb{E}_{S\sim D^{m-k+1}}\mathbb{E}_{(\mathbf{v}_1,\dots,\mathbf{v}_{k-1})\sim D^{k-1}}\left[\sup_{d\in\mathcal{G}_{\mathbf{v}_{1:k-1}}}\left|\frac{1}{m}\sum_{v\in S}d(v)+\frac{1}{m}\mathbb{E}_{(\mathbf{u}_1,\dots,\mathbf{u}_{k-1})\sim D^{k-1}}\sum d(\mathbf{u}_i)-\mathbb{E}_v\,d(v)\right.\right.$$

$$\left.\left.-\frac{1}{m}\mathbb{E}_{(\mathbf{u}_1,\dots,\mathbf{u}_k)\sim D^k}\sum d(\mathbf{u}_i)+\frac{1}{m}\sum_{i=1}^{k-1}d(\mathbf{v}_i)\right|\right]$$

$$\leq\mathbb{E}_{S\sim D^{m-k+1}}\mathbb{E}_{(\mathbf{v}_1,\dots,\mathbf{v}_{k-1})\sim D^{k-1}}\left[\sup_{d\in\mathcal{G}_{\mathbf{v}_{1:k-1}}}\left|\mathbb{E}_{(\mathbf{u}_1,\dots,\mathbf{u}_{k-1})\sim D^{k-1}}\left[\frac{1}{m}\sum_{v\in S\cup\{\mathbf{u}_1,\dots,\mathbf{u}_{k-1}\}}d(v)\right]-\mathbb{E}_v[d(v)]\right|\right.$$

$$\left.+\left|\frac{1}{m}\mathbb{E}_{(\mathbf{u}_1,\dots,\mathbf{u}_k)\sim D^k}\sum d(\mathbf{u}_i)-\frac{1}{m}\sum_{i=1)}^{k-1}d(\mathbf{v}_i)\right|\right]$$

$$\leq\mathbb{E}_{(\mathbf{u}_1,\dots,\mathbf{u}_k)\sim D^k}\left[\mathbb{E}_{S\sim D^{m-k}}\mathbb{E}_{(\mathbf{v}_1,\dots,\mathbf{v}_{k-1})\sim D^{k-1}}\left[\sup_{d\in\mathcal{G}_{\mathbf{v}_{1:k-1}}}\left|\frac{1}{m}\sum_{v\in S\cup\{\mathbf{u}_1,\dots,\mathbf{u}_k\}}d(v)-\mathbb{E}_v[d(v)]\right|\right]\right]+\frac{2k}{m}$$

437 Renaming $\mathbf{u}_1,\dots,\mathbf{u}_k$ and $\mathbf{v}_1,\dots,\mathbf{v}_k$ we can write:

$$\mathbb{E}_{(\mathbf{u}_1,\dots,\mathbf{u}_k)\sim D^k}\left[\mathbb{E}_{S\sim D^{m-k}}\mathbb{E}_{(\mathbf{v}_1,\dots,\mathbf{v}_{k-1})\sim D^{k-1}}\left[\sup_{d\in\mathcal{G}_{\mathbf{v}_{1:k-1}}}\left|\frac{1}{m}\sum_{v\in S/\cup\{\mathbf{u}_1,\dots,\mathbf{u}_k\}}d(v)-\mathbb{E}_v[d(v)]\right|\right]\right]+\frac{2k}{m}$$

$$=\mathbb{E}_{(\mathbf{v}_1,\dots,\mathbf{v}_k)\sim D^k}\left[\mathbb{E}_{S\sim D^{m-k}}\mathbb{E}_{(\mathbf{u}_1,\dots,\mathbf{u}_{k-1})\sim D^{k-1}}\left[\sup_{d\in\mathcal{G}_{\mathbf{u}_{1:k-1}}}\left|\frac{1}{m}\sum_{v\in S\cup(\mathbf{v}_1,\dots,\mathbf{v}_k)}d(v)-\mathbb{E}_v[d(v)]\right|\right]\right]+\frac{2k}{m}$$

$$=\mathbb{E}_{(\mathbf{u}_1,\dots,\mathbf{u}_{k-1})\sim D^{k-1}}\mathbb{E}_{S\sim D^m}\left[\sup_{d\in\mathcal{G}_{\mathbf{u}_{1:k-1}}}\left|\frac{1}{m}\sum_{v\in S}d(v)-\mathbb{E}_v[d(v)]\right|\right]+\frac{2k}{m}$$

Finally we apply. theorem 8. Recalling that $\mathrm{gVC}(\mathcal{D}_{\mathbf{u}_{1:k-1}}) = \rho$, and that the sequence $S$ is drawn IID independent of the choice $\mathbf{u}_{1:k-1}$, we obtain for every fixed $(\mathbf{u}_1, \ldots, \mathbf{u}_k)$

$$\mathbb{E}_{S \sim D^m} \left[ \sup_{d \in \mathcal{D}_{\mathbf{u}_{1:k-1}}} \left| \frac{1}{m} \sum_{v \in S} d(v) - \mathbb{E}_v[d(v)] \right| \right] \leq \frac{4 + \sqrt{\rho \log 2em/\rho}}{\sqrt{2m}}$$

**Bounding \*\***

$$\mathbb{E}_{S \sim D^m} \left[ \sup_{g \in \mathcal{G}} \left| \frac{1}{m^{k-1}} \sum_{\mathbf{v}_{1:k-1} \in S^{k-1}} \mathbb{E}_v \, g_{\mathbf{v}_{1:k-1}}(v) - \mathbb{E}_{\mathbf{v}_{1:k-1}} \mathbb{E}_v \, g_{\mathbf{v}_{1:k-1}}(v) \right| \right]$$

$$\leq \mathbb{E}_v \, \mathbb{E}_{S \sim D^m} \left[ \sup_{g \in \mathcal{G}} \left| \frac{1}{m^{k-1}} \sum_{\mathbf{v}_{1:k-1} \in S^{k-1}} g_{\mathbf{v}_{1:k-1}}(v) - \mathbb{E}_{\mathbf{v}_{1:k-1}} \, g_{\mathbf{v}_{1:k-1}}(v) \right| \right]$$

$$= \mathbb{E}_v \, \mathbb{E}_{S \sim D^m} \left[ \sup_{g \in \mathcal{G}} \left| \frac{1}{m^{k-1}} \sum_{\mathbf{v}_{1:k-1} \in S^{k-1}} g_v(\mathbf{v}_1, \ldots, \mathbf{v}_{k-1}) - \mathbb{E}_{\mathbf{v}_{1:k-1}} \, g_v(\mathbf{v}_1 \ldots, \mathbf{v}_{k-1}) \right| \right]$$

$$= \mathbb{E}_v \, \mathbb{E}_{S \sim D^m} \left[ \sup_{g \in \mathcal{G}_v} \left| \frac{1}{m^{k-1}} \sum_{\mathbf{v}_{1:k-1} \in S^m} g(\mathbf{v}_1, \ldots, \mathbf{v}_{k-1}) - \mathbb{E}_{\mathbf{v}_{1:k-1}} \, g(\mathbf{v}_1 \ldots, \mathbf{v}_{k-1}) \right| \right]$$

We now use the induction hypothesis: Note that $\mathcal{G}_v$ is $(k-1)$-distinguishing class with $\mathrm{gVC}(\mathcal{G}_v) = \rho$ for every choice of $v$. Thus, fixing $v$:

$$\mathbb{E}_{S \sim D^m} \left[ \sup_{g \in \mathcal{G}_v} \left| \frac{1}{m^{k-1}} \sum_{\mathbf{v}_{1:k-1} \in S^{k-1}} g(\mathbf{v}_1, \ldots, \mathbf{v}_{k-1}) - \mathbb{E}_{\mathbf{v}_{1:k-1}} \, g(\mathbf{v}_1 \ldots, \mathbf{v}_{k-1}) \right| \right]$$

$$\leq \frac{(k-1)\left( 4 + \sqrt{\rho \log(2em/\rho)} \right)}{\sqrt{2m}} + \frac{k(k-1)}{m}$$

**Continuing the proof**  With the aforementioned bound on the terms \* and \*\* we now obtain

$$* + ** \leq \frac{4 + \sqrt{\rho \log 2em/\rho}}{\sqrt{2m}} + \frac{2k}{m} + \frac{(k-1)\left( 4 + \sqrt{\rho \log(2em/\rho)} \right)}{\sqrt{2m}} + \frac{k(k-1)}{m}$$

$$= \frac{k\left( 4 + \sqrt{\rho \log(2em/\rho)} \right)}{\sqrt{2m}} + \frac{(k+1)k}{m}$$

## B.2  Proof of theorem 3

**Theorem 3** (Sample Complexity – Lower Bound)**.** *Let $\mathcal{G}$ be a $k$–distinguishing class with $\mathrm{gVC}(\mathcal{G}) = \rho$. Any $\mathcal{G}$-discriminating algorithm with accuracy $\epsilon > 0$ that succeeds with probability $1 - \frac{2^{-k \log k}}{3}$, must observe at least $\Omega\left( \frac{\sqrt{\rho}}{2^{7k^3} \epsilon^2} \right)$ samples.*

To prove theorem 3 we will in fact prove a stronger statement: We will show that it is not only hard to compute a $g \in \mathcal{G}$ as required, but in fact it is even hard to determine if such $g$ exists vs. the case that $p_1 = p_2$.

Specifically let us call an algorithm $A$ a testing algorithm for $\mathcal{G}$ with sample complexity $m(\epsilon, \delta)$ if $A$ receives IID samples from two distributions $p_1$ and $p_2$ of size $m(\epsilon, \delta)$ and returns either $EQUIVALENT$ or $DISTINCT$ such that w.p. $(1 - \delta)$:

- If $p_1 = p_2$ the algorithm returns $EQUIVALENT$.

454     • If $\text{IPM}_{\mathcal{G}}(p_1, p_2) > \epsilon$ the algorithm returns $DISTINCT$

455 **Theorem 10.** *Let $\mathcal{G}$ be a $k$–distinguishing class with $\text{gVC}(\mathcal{G}) = \rho$. Any testing algorithm $A$ with*

456 *sample complexity $m(\epsilon, \delta)$ must observe $\Omega\left(\frac{\sqrt{\rho}}{2^{7k^3}\epsilon^2}\right)$ examples for any $\delta < \frac{2^{-k\log k}}{3}$.*

457 Clearly, theorem 3 is a corollary of theorem 10. Indeed if $A$ is a discriminating algorithm for $\mathcal{G}$ with

458 sample complexity $m(\epsilon, \delta)$ we can apply it over a sample of size $m(\epsilon/3, \delta)$ to receive (w.p. $1 - \delta$) a

459 graph $g$ s.t.

$$\text{IPM}_{\mathcal{G}}(p_1, p_2) \leq |\mathbb{E}_{p_1}(g) - \mathbb{E}_{p_2}(g)| + \frac{\epsilon}{3}.$$

460 With an additional sample of size $O(\frac{\log 1/\delta}{\epsilon^2})$ we can estimate $|\mathbb{E}_{p_1}(g) - \mathbb{E}_{p_2}(g)|$ within accuracy $\epsilon/3$,

461 and verify if $\text{IPM}_{\mathcal{G}}(p_1, p_2) < \epsilon$: The test will then output $EQUIVALENT$ if $|\mathbb{E}_{p_1}(g) - \mathbb{E}_{p_2}(g)| <$

462 $\frac{\epsilon}{3}$. It thus follows that, for sufficiently small $\delta$ $m(\epsilon, \delta) > \Omega(\frac{\sqrt{\rho}}{2^{7k^3}\epsilon^2})$.

463 We proceed with the proof of theorem 10.

### B.2.1 Proof of theorem 10

465 The proof is done by induction. For the induction, we will assume a more fine-grained lower bound.

466 We will assume that there exists a constant $C$ so that for every $n \leq k - 1$, if $m_n(\epsilon, \delta)$ is the sample

467 complexity of a $n$-distinguishing class then:

$$m_n(\epsilon, \delta) \geq C \frac{\sqrt{\rho}}{(n+1)! 2^{\sum_{j=1}^n 6j^2} \cdot \epsilon^2} = \Omega\left(\frac{\sqrt{\rho}}{2^{7n^3}\epsilon^2}\right). \tag{5}$$

468 $C > 0$ will depend only on the constant for the lower bound for testing if two distributions are distinct

469 or $\epsilon$-far in total variation, as in theorem 9.

470 We start with the case $k = 1$.

471 $\underline{k = 1}$ The case $k = 1$ follows directly from theorem 9. Let $\mathcal{D}$ be a class with VC dimension $\rho$. by

472 restricting our attention to probabilities supported on the shattered set of size $\rho$, we may assume that

473 $|\mathcal{V}| = \rho$ and that $\mathcal{D} = P(\mathcal{V})$. Note then, that for the IPM distance we then have

$$\text{IPM}_{\mathcal{D}}(p_1, p_2) = \text{TV}(p_1, p_2).$$

474 theorem 9 immediately yields the result.

475 the induction step We now proceed with the proof assuming the statement holds for $k - 1$.

476 By assumption $\text{gVC}(\mathcal{G}) = \rho$. Fix $v \in \mathcal{V}$ such that $\text{gVC}(\mathcal{G}_v) = \rho$. For every $q \in (0, 1)$ and

477 distribution $p$ denote

$$p^q := q\delta_v + (1 - q)p. \tag{6}$$

478 We next state the core Lemma we will need for the proof:

479 **Lemma 2.** *Let $\mathcal{G}$ be a family of $k$-hypergraphs and $p_1, p_2$ two distributions. Assume that for some*

480 *$v \in \mathcal{V}$ we have that:*

$$\text{IPM}_{\mathcal{G}_v}(p_1, p_2) \geq \epsilon.$$

481 *Let $p_1^q$ and $p_2^q$ be as in eq. (6) for our choice of $v \in \mathcal{V}$.*

482 *Then for some value $q \in \{0, \frac{1}{k}, \frac{2}{k}, \cdots 1\}$ we have that,*

$$\text{IPM}_{\mathcal{G}}(p_1^q, p_2^q) \geq \frac{\epsilon}{2^{3k^2}}.$$

483 We deter the proof of lemma 2 to appendix B.2.2, and proceed with the proof of the induction step.

484 Let us denote $\delta_k = 2^{-k\log k}$ and denote $c_k = 2^{-3k^2}$.

485 Let $A$ be a testing algorithm for $\mathcal{G}$ with sample complexity $m(\epsilon, \delta)$ as in theorem 10. We can now

486 construct a testing algorithm for $\mathcal{G}_v$ with sample complexity

$$m_{k-1}(\epsilon, \delta) = (k + 1) \cdot m(c_k \epsilon, \frac{\delta}{k}),$$

487 as follows: Run the testing algorithm $A$ on pairs of distributions $(p_1, p_2), (p_1^{1/k}, p_2^{1/k}), \ldots, (p_1^1, p_2^1)$,
488 each on its own fixed sample of size $m(c_k\epsilon, \frac{\delta}{k})$. If the algorithm returns $DISTINCT$ for any of
489 these tests, output $DISTINCT$, else output $EQUIVALENT$.

490 We now show that if $p_1 = p_2$ the algorithm outputs w.p. $(1-\delta)$ $EQUIVALENT$: Indeed, since
491 $p_1 = p_2$, we have that $p_1^q = p_2^q$ for all $q$: Applying union bound we have that w.p. $(1-\delta)$ the
492 algorithm indeed outputs $EQUIVALENT$.

493 On the other hand, if $\mathrm{IPM}_{\mathcal{G}_v}(p_1, p_2) \geq \epsilon$ we have by lemma 2 that for one of the distributions
494 $(p_1^q, p_2^q)$, $\mathrm{IPM}_{\mathcal{G}}(p_1^q, p_2^q) > c_k\epsilon$, in particular the algorithm will output $DISTINCT$ with probability
495 $(1-\delta)$. Overall we constructed a testing algorithm for $\mathcal{G}_v$ with sample complexity $(k+1)m(c_k\epsilon, \frac{\delta}{k})$

496 Since $\delta_k < \frac{2^{-(k-1)\log(k-1)}}{k}$, it follows from the induction step

$$(k+1)m(c_k\epsilon, \frac{\delta}{k}) = m_{k-1}(\epsilon, \delta)$$

$$\geq C\frac{\sqrt{\rho}}{k!2^{\sum_{n=1}^{k-1}6n^2} \cdot \epsilon^2}$$

497 Reparametrizing we obtain

$$m(\epsilon, \frac{\delta}{k}) \geq C\frac{\sqrt{\rho}}{(k+1)!2^{\sum_{n=1}^{k}6n^2} \cdot \epsilon^2}$$

498 **B.2.2   Proof of lemma 2**

499 Denote
$$\Delta_n^g(p_1, p_2) = \mathop{\mathbb{E}}_{\mathbf{u}_{1:n} \sim p_1^{k-n}} g(\underbrace{v, v, v, \ldots, v}_{n \text{ times}}, \mathbf{u}_1, \ldots, \mathbf{u}_{k-n}) - \mathop{\mathbb{E}}_{\mathbf{u}_{1:n} \sim p_2^{k-n}} g(\underbrace{v, v, v, \ldots, v}_{n \text{ times}}, \mathbf{u}_1, \ldots, \mathbf{u}_{k-n})$$

500 One can show that
$$\mathrm{IPM}_{\mathcal{G}}(p_1^q, p_2^q) = \sup_{g \in \mathcal{G}} \left| \sum \binom{k}{n} q^n (1-q)^{k-n} \Delta_n^g(p_1, p_2) \right|$$

$$= \sup_{g \in \mathcal{G}} \left| (1-q)^k \Delta_0^g(p_1, p_2) + kq(1-q)^{n-1}\Delta_1^g(p_1, p_2) + \sum_{n=2}^{k} \binom{k}{n} q^n (1-q)^{k-n}\Delta_n^g(p_1, p_2) \right|$$

$$= \sup_{g \in \mathcal{G}} \left| \Delta_0^g(p_1, p_2) + kq\left(\Delta_1^g(p_1, p_2) - \Delta_0^g(p_1, p_2)\right) + q^2 p_g(q) \right|$$

501 where $p_g(q)$ is some $k-2$ degree polynomial in $q$ whose coefficient depend on $g$ and $p_1$ and $p_2$. We
502 next apply the following claim

503 **Claim 1.** *Let* $f(q) = a_0 + a_1 q + q^2 p(q)$ *where* $p(q)$ *is some* $k-2$ *degree polynomial. then for some*
504 *value* $q_0 \in \{0, \frac{1}{k}, \frac{2}{k}, \cdots 1\}$ *we have that* $|f(q_0)| \geq \frac{|a_1|}{2^{3k^2}}$

505 *Proof Sketch.* We provide a full proof for this claim in appendix D.1. In a nutshell, claim 1 follows
506 from the equivalence between norms in finite dimensional spaces. Indeed, the mapping
$$(a_0, \ldots, a_k) \to (p_a(1/k), p_a(2/k), \ldots, p_a(1)),$$
507 where $p_a(x) = \sum a_i x^i$ is known to be a non–singular linear transformation induced by the appropri-
508 ate Vandermonde matrix (specifically. $V_{i,j} = ((i-1)/k))^{j-1}$). Letting $\lambda_{min}$ be the smallest singular
509 value of the matrix $V$, we know that $\|V\mathbf{a}\|_2 \geq \lambda_{min}\|\mathbf{a}\|_2$. where $\mathbf{a}$ is the vector of coefficients of the
510 polynomial $p_a$.

511 Finally, we exploit the relation in $\mathbb{R}^{k+1}$: $\|x\|_\infty \leq \|x\|_2 \leq \sqrt{k+1}\|x\|_\infty$. We can, thus, relate the
512 max norm of the coefficient vector $\|a\|_\infty \geq |a_1|$ to the maximum value $\max_{i\in\{0,\ldots,k\}} \sum a_j(i/k)^j =$
513 $\|V\mathbf{a}\|_\infty$ to obtain

$$|a_1| \leq \|\mathbf{a}\|_2 \leq \lambda_{min}^{-1}\|V\mathbf{a}\|_2 \leq \frac{\sqrt{k+1}}{\lambda_{min}}\|V\mathbf{a}\|_\infty = \frac{\sqrt{k+1}}{\lambda_{min}} \max_{i\in\{0,\ldots,k\}} \sum a_j(i/k)^j$$

514 It remains only to lower bound the singular values of $V$, this is done in the full proof in appendix D.1.

515 □

516 With claim 1 in mind we prove the result as follows: First, suppose that for some $g \in \mathcal{G}$ we have
517 that $|k(\Delta_0^g(p_1, p_2) - \Delta_1^g(p_1, p_2)| > \frac{\epsilon}{2}$. In this case, applying claim 1 with $a_0 = \Delta_0^g(p_1, p_2)$ and
518 $a_1 = k(\Delta_0^g(p_1, p_2) - \Delta_1^g(p_1, p_2))$ and $p = p_g$, we obtain that there exists a value $q = j/k$ such that
519 $\mathrm{IPM}_{\mathcal{G}}(p_1^q, p_2^q) \geq \frac{\epsilon}{2^{3k^2}}$.

520 On the other hand, consider the case that $|k(\Delta_0^g(p_1, p_2) - \Delta_1^g(p_1, p_2)| < \frac{\epsilon}{2}$ for any $g \in \mathcal{G}$, by
521 assumption we have that $|\Delta_1^g(p_1, p_2)| > \epsilon$, for some $g \in \mathcal{G}$. Hence $|\Delta_0^g(p_1, p_2)| > \epsilon$. By definition
522 of $\Delta_0$ we have that for $q = 0$ we obtain that: $\mathrm{IPM}_G(p_1^q, p_2^q) = |\mathbb{E}(p_1^q) - \mathbb{E}(p_2^q)| > \frac{\epsilon}{2}$.

## C  Expressivity – Proofs

### C.1  Proof of theorem 4

525 **Theorem 4.** *Let $\mathcal{V} = \mathbb{N}$. There exists a distinguishing graph class $\mathcal{G}$, with sample complexity*
526 $m(\epsilon, \delta) = O(\frac{\log 1/\delta}{\epsilon^2})$ *(in fact $|\mathcal{G}| = 1$) such that: for any 1-distinguishing class $\mathcal{D}$ with finite VC*
527 *dimension, and every $\epsilon > 0$ there are two distributions $p_1, p_2$ such that $\mathrm{IPM}_{\mathcal{D}}(p_1, p_2) < \epsilon$ but*
528 $\mathrm{IPM}_{\mathcal{G}}(p_1, p_2) > 1/2$

529 As stated, the class $\mathcal{G}$ will consist of a single graph $g$. The graph $g$ is going to be a bipartite
530 graph. We thus, divide the vertices into two infinite sets: $\mathcal{V}_1$ and $\mathcal{V}_2$ the elements of $\mathcal{V}_1$ will be
531 indexed by $\mathbb{N}$ i.e. $\mathcal{V}_1 = \{v_1, v_2, \cdots\}$ and we index the elements of $\mathcal{V}_2$ with finite subsets of $\mathbb{N}$
532 $V_2 = \{v_A : A \subseteq \mathbb{N}, |A| < \infty\}$. Next we define $g$ so that an edge passes between $v_i \in \mathcal{V}_1$ and
533 $v_A \in \mathcal{V}_2$ iff $i \in A$.

534 Let $\mathcal{D}$ be a distinguishing class with finite sample complexity, in particular $\mathrm{gVC}(\mathcal{D}) < \infty$. Denote
535 $\mathrm{gVC}(\mathcal{D}) = \rho$. Let $\mathcal{D}_1$ be the restriction of $\mathcal{D}$ to $\mathcal{V}_1$: Note that $\mathrm{gVC}(\mathcal{D}_1) \leq \rho$.

536 Next we make the following claim:

537 **Claim 2.** *There are two distributions, $q_1$ and $q_2$, supported on $\mathcal{V}_1$ so that*

$$\mathrm{IPM}_{\mathcal{D}_1}(p_1, p_2) < \epsilon.$$

538 *and yet $q_1$ and $q_2$ have disjoint support.*

539 *Proof.* To construct two such distributions, choose a set $S \subseteq \mathcal{V}_1$ of size $m$ large enough (to be
540 determined later). Then, randomly choose two samples $S_1$ and $S_2$ out of $S$ (uniformly), each of size
541 $O(\frac{\rho}{\epsilon^2})$. Then, by theorem 2 with some constant probability we have that $\mathrm{IPM}_{\mathcal{D}}(p_{S_1}, p_S) < \epsilon/2$ and
542 similarly $\mathrm{IPM}_{\mathcal{D}}(p_S, p_{S_2}) < \epsilon/2$ . Taken together we obtain that $\mathrm{IPM}_{\mathcal{G}}(p_{S_1}, p_{S_2}) < \epsilon$.

543 Also, if $S$ is sufficiently large (say, of order $O(\frac{\rho^2}{\epsilon^4})$), we would have that w.h.p $S_1 \cap S_2 = \emptyset$. Thus, let
544 $q_1 = p_{S_1}$ and $q_2 = p_{S_2}$. □

545 With claim 2, we proceed with the proof. Let $q_1$ and $q_2$ be as in claim 2. Let $A$ be the support of $q_1$,
546 and define $p_1$ to be a distribution $p_1 = \frac{1}{2}\delta_A + \frac{1}{2}q_1$ and similarly we define $p_2 = \frac{1}{2}\delta_A + \frac{1}{2}q_2$. We then
547 have

$$\mathrm{IPM}_{\mathcal{D}}(p_1, p_2) = \frac{1}{2}\mathrm{IPM}_{\mathcal{D}}(q_1, q_2)$$
$$= \frac{1}{2}\mathrm{IPM}_{\mathcal{D}_1}(q_1, q_2)$$
$$< \epsilon.$$

548 On the other hand, note that for $p_1$ the probability to draw an edge from $g$ is at least $1/2$ (indeed if
549 $v_1 = v_A$ and $v_2 \neq v_A$ drawn from $q_1$ then $g(v_1, v_2) = 1$. On the other hand, the probability to draw
550 an edge from $p_2$ is 0. It follows that

$$\mathrm{IPM}_{\mathcal{G}}(p_1, p_2) > \frac{1}{2}.$$

## C.2 Proof of theorem 5

**Theorem 5.** *Let $\mathcal{V} = \mathbb{N}$. There exists a $k$-distinguishing class $\mathcal{G}_k$, with sample complexity $m(\epsilon, \delta) = O(\frac{k^2 + \log 1/\delta}{\epsilon^2})$ such that: For any $k - 1$-distinguishing class $\mathcal{G}_{k-1}$ with bounded sample complexity, and every $\epsilon > 0$ there are two distributions $p_1, p_2$ such that $\mathrm{IPM}_{\mathcal{G}_{k-1}}(p_1, p_2) < \epsilon$ and $\mathrm{IPM}_{\mathcal{G}_k}(p_1, p_2) > 1/4$.*

The construction is similar to the case $k = 2$. We again divide the vertices into two infinite sets $\mathcal{V}_1$ and $\mathcal{V}_2$. Again, the elements of $\mathcal{V}_1$ will be indexed by $\mathbb{N}$, and the elements of $\mathcal{V}_2$ are indexed by finite subsets of $\mathbb{N}$. $\mathcal{V}_2 = \{v_A : A \subseteq \mathbb{N}, |A| < \infty\}$.

We define the hyper graph $g_k$ to be a (undirected) graph that contains a hyperedge $(v_{i_1}, \ldots, v_{i_{k-1}}, v_A)$ whenever $\{i_1 \ldots, i_{k-1}\} \subseteq A$.

Next, as before we construct two distributions with distinct support such that $\mathrm{IPM}_{\mathcal{G}}(p_1, p_2) \leq \epsilon$. This is done similar to the proof of theorem 4. Specifically:

**Claim 3.** *Let $\mathcal{G}$ be a $k - 1$-distinguishing class defined on $\mathcal{V}_1$. There are two distributions, $q_1$ and $q_2$, supported on $\mathcal{V}_1$ so that*

$$\mathrm{IPM}_{\mathcal{G}}(p_1, p_2) < \epsilon.$$

*and yet $q_1$ and $q_2$ have disjoint support.*

The proof is a repetition of the proof of claim 2, where we draw $S_1$ and $S_2$ to be order of $O(\frac{k^2 \rho}{\epsilon^2})$, and again invoke theorem 2.

As before, then, given a class $\mathcal{G}$ of $k - 1$–hypergraphs we take two distributions $q_1$ and $q_2$ as in claim 3 and if $A$ is the support of $q_1$, we take $p_1 = \frac{1}{k}\delta_{v_A} + (1 - \frac{1}{k})q_1$ and let $p_2 = \frac{1}{k}\delta_{v_A} + (1 - \frac{1}{k})q_2$. Then, we can show that $\mathrm{IPM}_{\mathcal{G}}(p_1, p_2) \leq \epsilon$. On the other hand, the probability to draw an edge from $g_k$ is $k \cdot \frac{1}{k}(1 - \frac{1}{k})^{k-1} \geq e^{-1}$ according to $p_1$, but the probability to draw an edge from $p_2$ is 0.

## C.3 Proof of theorem 6

**Theorem 6.** *Let $|\mathcal{V}| = n$. There exists a $k$-distinguishing class $\mathcal{G}_k$, with sample complexity $m(\epsilon, \delta) = O(\frac{k^2 + \log 1/\delta}{\epsilon^2})$ (in fact $|\mathcal{G}| = 1$) such that: For any $\epsilon > 0$ and any $k - 1$ distinguishing class $\mathcal{G}_{k-1}$ if:*

$$\mathrm{IPM}_{\mathcal{G}_{k-1}} \succ \epsilon \cdot \mathrm{IPM}\mathcal{G}_k$$

*then $\mathrm{gVC}(\mathcal{G}_{k-1}) = \Omega(\frac{\epsilon^2}{k^2}\sqrt{\log n})$.*

The proof is similar to the proof of theorem 5. For simplicity, let us assume that $|\mathcal{V}| = n + \log n$. This will not change the results up to constants.

Given $n + \log n$ vertices we partition them into two sets $\mathcal{V}_1$, of size $\log n$ and $\mathcal{V}_2$. We index the elements of $\mathcal{V}_1$ as $\{v_1, \ldots, v_{\log n}\}$ and we index the elements of $\mathcal{V}_2$ with subsets of $[\log n]$. We then consider a graph $g$ that contains only hyper-edges of the form $(v_{i_1}, \ldots, v_{k-1}, v_A)$ iff $\{i_1, \ldots, i_{k-1}\} \in A$.

Next, let $\mathcal{G}_{k-1}$ be a distinguishing class with $\mathrm{gVC}(\mathcal{G}_{k-1}) = \rho$, and let $m(\epsilon, \delta) = O\left(\frac{\rho k^2}{\epsilon^2}\right)$ be an upper bound on the sample complexity of classes of graph VC dimension $\rho$.

We claim that if $\log n \geq m^2(\epsilon/8, 0.99)$ then there are two distinct distributions $q_1, q_2$ over $[\log n]$, with disjoint support such that $\mathrm{IPM}_{\mathcal{G}_k}(q_1, q_2) < \epsilon$. The proof is done as in claim 3.

Indeed, we draw IID, and uniformly, two random samples $S_1$ and $S_2$ from $\{1, \ldots, \log n\}$ of size $m(\epsilon/8, 0.99)$. One can show that w.p $1/4$ we have that $S_1 \cap S_2$ are distinct, also we have w.p 0.98 that $\mathrm{IPM}_{\mathcal{G}}(p_S, p_{S_1}) < \epsilon/8$ and similarly $\mathrm{IPM}_{\mathcal{G}}(p_S, p_{S_2}) < \epsilon/8$. Taken together we obtain that with positive probability $q_1 = p_{S_1}$ and $q_2 = p_{S_2}$ have disjoint support and $\mathrm{IPM}_{\mathcal{G}}(q_1, q_2) < \frac{\epsilon}{4}$.

As in theorem 5, let $A$ be the support of $q_1$ and consider a distribution $p_1 = \frac{1}{k}\delta_{v_A} + (1 - \frac{1}{k})q_1$ and similarly $p_2 = \frac{1}{k}\delta_{v_A} + (1 - \frac{1}{k}q_2$. One can show that $\mathrm{IPM}_{\mathcal{G}}(p_1, p_2) < \frac{\epsilon}{4}$ but the probability to draw an edge from $g$ according to $q_1$ is at least $1/4$, while it equals 0 if we draw edges according to $p_2$.

593 To conclude, we showed that if $\log n \geq m^2(\epsilon/8, 0.99)$ then $\text{IPM}_{\mathcal{G}_k} \prec \epsilon\text{IPM}_{\mathcal{G}}$. In other words, if
594 $\text{IPM}_{\mathcal{G}_k} \succ \epsilon \cdot \text{IPM}_{\mathcal{G}}$ then $\log n \leq m^2(\epsilon/8, 0.99)$.

$$\rho = \Omega\left(\frac{\epsilon^2}{k^2}\sqrt{\log n}\right).$$

## C.4 Proof of theorem 7

595

596 **Theorem 7.** *Let $|\mathcal{V}| = n$. There exists a 2-distinguishing class $\mathcal{G}$, with sample complexity $m(\epsilon, \delta) =$
597 $O(\frac{\log 1/\delta}{\epsilon^2})$ (in fact $|\mathcal{G}| = 1$) such that: For any $\epsilon > 0$ and any distinguishing class $\mathcal{D}$ if:*

$$\text{IPM}_{\mathcal{D}} \succ \epsilon \cdot \text{IPM}\mathcal{G}$$

598 *then $\text{gVC}(\mathcal{D}) = \tilde{\Omega}(\epsilon^2 \log n)$.*

599 The proof is similar to the proof of theorem 4 but we will use an improved upper bound on the size of
600 $S$ which we next state (see appendix C.5 for a proof):

601 **Lemma 3.** *Let $\mathcal{D}$ be a class with $\text{gVC}(\mathcal{D}) = \rho$ over a domain $S$. There exists a constant $c > 0$
602 (independent of $\mathcal{D}$ and $d$) such that if $|S| > c \cdot \frac{d}{\epsilon^2} \log^2(d/\epsilon^2)$, Then there are two distributions $q_1$ and
603 $q_2$, supported on $S$ such that:*

604     *1. $q_1$, and $q_2$ have disjoint support.*

605     *2. $\text{IPM}_{\mathcal{D}}(q_1, q_2) < \epsilon$*

606 The graph $g$ is constructed as in theorem 4. Let $\mathcal{V}$ be a set of vertices of size $n + \log n$, let $\mathcal{V}_1$ be a
607 set of size $\log n$ and we index its elements with $\{v_1, \ldots, v_2, \ldots, v_{\log n}\}$. We let $\mathcal{V}_2$ include all other
608 elements and we index them via subsets of $[\log n]$. The graph is again constructed so that $v_A \in \mathcal{V}_2$
609 has an edge to $v_i \in \mathcal{V}_1$ iff $i \in A$. As before, we make the graph bipartite, i.e. both $\mathcal{V}_1$ and $\mathcal{V}_2$ are
610 independent sets.

611 Now suppose $\log n \geq c\frac{\rho}{\epsilon^2} \log^2 \frac{d}{\epsilon^2}$. By lemma 3 we have that there exists a set $A \subseteq \{1, \ldots, \log n\}$, a
612 distribution $p_1$ and $p_2$ where $p_1$ is supported on $A$ and $p_2$ is supported on its compelement so that
613 $\text{IPM}_{\mathcal{G}}(p_1, p_2) < \epsilon$. As before we construct $q_1 = \delta v_A + (1 - \delta)p_1$ and $q_2 = \delta v_A + (1 - \delta)p_2$. One
614 can verify that $\text{IPM}_{\mathcal{G}}(q_1, q_2) < \epsilon$ but $\text{IPM}_{\mathcal{G}_{k+1}}(q_1, q_2) > \frac{1}{2}$. Thus, if $\text{IPM}_{\mathcal{G}_k} \succ \epsilon \cdot \text{IPM}_{\mathcal{G}_{k+1}}$ then
615 $\log n \leq c\frac{\rho}{\epsilon^2} \log^2 \frac{d}{\epsilon^2}$. In turn $d = \tilde{\Omega}(\epsilon^2 \log n)$.

## C.5 Proof of lemma 3

616

617 First w.l.o.g we assume that the constant functions are in $\mathcal{D}$ (i.e. 0 and 1).

618 We want to choose a constant $c$ so that if $|S| \geq c\frac{2d}{\epsilon^2} \log^2 \frac{2d}{\epsilon^2}$, then we have $\frac{|S|}{\ln^2 |S|} > \frac{2\rho \log e}{\epsilon^2}$. Fix such
619 $c > 0$, and let $\mathcal{H}_m = \{\text{sign}(\sum_{i=1}^m (2d_i(v) - 1)) : d_i \in \mathcal{D}\}$ and denote $\mathcal{H} = \mathcal{H}_{\frac{2}{\epsilon^2} \ln |S|}$. Note that

$$
\begin{aligned}
|\mathcal{H}| &\leq |\mathcal{D}|^{\frac{2}{\epsilon^2} \ln |S|} \\
&\leq |S|^{\frac{2\rho}{\epsilon^2} \ln |S|} \qquad\qquad \text{Sauer's Lemma} \\
&= 2^{\frac{2\rho \log e}{\epsilon^2} \ln^2 |S|} \\
&< 2^{|S|}
\end{aligned}
$$

620 It thus follows that there exists $f \notin \mathcal{M}$. Let $f$ be such and define a matrix $M = \{0, 1\}^{|S| \times |\mathcal{D}|}$ so that

$$M_{v,d} = \begin{cases} 1 & d(v) \neq f(v) \\ 0 & \text{else} \end{cases}$$

Now suppose that for some distribution $q$ over $S$, for every $d$ we have that $\mathbb{E}_{v \sim q}[d(v) = f(v)] < \frac{1}{2} + \frac{1}{\epsilon}$. Then, defining $q_1 = q(\cdot|f(v) = 0)$ and $q_2 = q(\cdot|f(v) = 1)$ yields the desired result. Indeed,

$$
\begin{aligned}
\sup_{d \in \mathcal{D}} |\mathbb{E}_{q_1}[d] - \mathbb{E}_{q_2}[d]| &= \sup_{d \in \mathcal{D}} 2|\frac{1}{2}\mathbb{E}_{q_1}[d] - \frac{1}{2}\mathbb{E}_{q_2}[d]| \\
&\geq \sup_{d \in \mathcal{D}} 2|q(f(v) = 1)\mathbb{E}_{q_1}[d] - q(f(v) = -1)\mathbb{E}_{q_2}[d]| - 4\max_{y \in \{1,-1\}}\{|\frac{1}{2} - q(f(v) = y)| \\
&\quad \sup_{d \in \mathcal{D}} 2|\mathbb{E}_{(v,y) \sim q} yd(v)| - 4\epsilon \\
&= \sup_{d \in \mathcal{D}} 2|1 - 2q(d(v) \neq f(v))| - 4\epsilon \\
&\geq 8\epsilon.
\end{aligned}
$$

We now wish to prove that indeed, such a $q$ exists. Suppose, otherwise: That for any distribution $q$ over $S$ we can find $d$ such that $\mathbb{E}_{v \sim q}[d(v) = f(v)] > \frac{1}{2} + \frac{1}{\epsilon}$. This can be rephrased in terms of a value of a minimax game as follows:

$$
\max_{q \in \Delta(S)} \min_{d \in \mathcal{D}} q^\top M_d < \frac{1}{2} - \epsilon,
$$

Where $\Delta(S)$ denotes the set of distributions over $S$. It is well known ([16], thm 2), that for any game defined by any matrix $M$ with $c$ columns, there exists a strategy for the row player that chooses uniformly from a multiset of $\frac{\ln c}{2\epsilon^2}$ and achieves $\epsilon$-optimiality.

In our setting, this translate to a uniform distribution $p$, supported on $\frac{\ln |S|}{2\epsilon^2}$ distinguishers $\{d_1, \ldots, d_{\frac{\ln |S|}{2\epsilon^2}}\}$ such that

$$
\frac{2\epsilon^2}{\ln |S|} \sum_{d_i} [d_i(v) \neq f(v)] < \frac{1}{2}
$$

, this contradicts the fact that $f \notin \mathcal{M}$.

We thus obtain that there exists a distribution $q$ over $S$ so that for every $d \in \mathcal{D} \sum q(v)[d(v) \neq f(v)] > \frac{1}{2} - \epsilon$.

# D   Additional Proofs

## D.1   Proof of claim 1

Consider the Vandermonde Matrix $V \in M_{k+1,k+1}$ given by $V_{i,j} = \left(\frac{i-1}{k}\right)^{j-1}$. Our first step will be to lower bound the smallest singular value of $V$. In turn, we will obtain a lower bound on the maximum value over the coordinates of the vector $V\mathbf{a}$. The proof can then be derived from the identity: $(V\mathbf{a})_i = \sum_{j=1}^{k+1} a_j \left(\frac{i-1}{k}\right)^j$.

Let $\lambda_1 \leq \lambda_2 \leq \ldots \leq \lambda_{k+1}$ be the singular values of $V$. To bound the smallest singular value, $\lambda_1$, we first observe that $\lambda_{k+1}$– the highest singular value is bounded by $k + 1$. To see that $\lambda_{k+1} \leq k + 1$, observe that for any vector $\|\mathbf{a}\| \leq 1$ we have that

$$
\|V\mathbf{a}\|_2 \leq k + 1 \max |V_{i,j}||a_i| \leq k + 1.
$$

Next, using the formula for the determinant of a Vandermonde matrix, and the relation $\det(V) = \prod \lambda_i$, we obtain:

$$
\begin{aligned}
\prod_{i=1}^{k+1} |\lambda_i| &= |\det(V)| \\
&= \prod_{1 \leq i < j \leq k+1} \frac{|i - j|}{k} \\
&\geq 2^{-\frac{k(k-1)\log k}{2}}
\end{aligned}
$$

645    Taken together we obtain

$$
\begin{aligned}
\lambda_{min} &\geq \frac{2^{-\frac{k(k-1)\log k}{2}}}{\prod_{i=2}^{k+1}\lambda_i} \\
&\geq \frac{2^{-\frac{k(k-1)\log k}{2}}}{\lambda_{k+1}^k} \\
&\geq 2^{-k(k-1)\log k - k\log(k+1)} \\
&= 2^{-k^2 + k\log k / k + 1} \\
&\geq 2^{-2k^2}
\end{aligned}
$$

646    Finally, for any polynomial $p = \sum a_i q^i$ with coefficient $|a_1|$ we have that $\|\mathbf{a}\|_2 \geq |a_1|$. We thus
647    obtain,

$$
\begin{aligned}
\max_i p(\frac{i}{q}) &\geq \frac{1}{\sqrt{k+1}}\sqrt{\sum |p(\frac{i}{q})|^2} \\
&= \frac{1}{\sqrt{k+1}}\|V\mathbf{a}\|_2 \\
&\geq \frac{1}{\sqrt{k+1}}\lambda_1\|\mathbf{a}\|_2 \\
&\geq \frac{2^{-2k^2 - 1/2\log(k+1)}}{k+1}|a|_1 \\
&\geq 2^{-3k^2}|a|_1
\end{aligned}
$$