[Reviews · NeurIPS 2019]

Reviewer 1



This is a good paper. The idea of using multivariate functions in order to perform better learning (distinguishing different distributions) is quite evident, especially when considering the fact that the assessment of the quality of the discrimination in the paper is expectation-related. It is an interesting and well structured paper. The statistical analysis of the Graph-based Discriminator classes is clear and informative and invites the follow up questions presented (closing down the gap in sample complexity, extending the expressiveness bounds for larger k). A few other points regarding this paper: 1) In line 94 - You state that o(\rho) examples are sufficient for discrimination. I believe (even though this is early in the paper) that it would be clearer to state this size using the \epsilon and \delta confidence and accuracy parameters at this stage. 2) In line 229 - You write 'fix all coordinates'. It is possible that this phrase is ambiguous but my understanding is that you intend to fix only a single coordinate while all others remain parameters of the function (executing a reduction to k-1 gVC), While what I understand from the reading the phrase is that after fixing all but one coordinate we end up with a function which receives 1 parameter instead of k-1. 3) I would be very interested in seeing future work regarding different types of discrimination 'loss' (equation 1 in line 31) since the difference in expectations of distinguishing functions hardly seems the best tool for differentiating distributions using a minimal amount of tests. Best of luck

Reviewer 2



Originality: The paper introduces a new notion of VC-dimension for k-ary Boolean functions that controls the sample complexity of graph-based discriminators. I do not think that such a notion of VC-dimension has been considered in the literature. Quality: The main results and the proofs generally seem to be correct. All the main claims in the paper are supported with complete proofs or references to proofs in the literature. I did not check all the proofs in detail; I read the proofs of Theorem 2 and Lemma 1 and they seem to be correct (but please check for typos; some of them are mentioned below under "Minor Comments"). Clarity: I was a bit confused by the definition of the empirical frequency of an edge; please refer to the General Comment/Question below. I also recommend that the paper (including the supplementary material) be proofread at least once more for typos; please see the "Minor Comments" below. Significance: I think the results in the paper are quite significant from an information-theoretic viewpoint of learning. For practical applications, one issue that I think was not discussed (as this was probably not the focus of the paper) is the computation of the IPM; although higher order distinguishing classes have more distinguishing power, how does the complexity of estimating the IPM w.r.t. these classes scale with the order? General Comment/Question: The definition of the empirical frequency of an edge (between lines 176 and 177 on page 5) seems a bit unusual if I am interpreting it correctly (if I am reading the equation wrongly, perhaps the definition could be clarified just before line 177?). The expression |{(u_1,...,u_k) \in E_g: \forall i,u_i \in S}| seems to count the number of distinct k-tuples chosen from S^k that belong to E_g. Why are multiple occurrences of the same k-tuple in S^k not counted separately? Counting them separately would seem to encode more information about the distribution on the vertices...? As an example, suppose S = {v_1,v_2,v_3} with v_1 = v_2 and v_1 \neq v_3, k = 1, and E_g = {{v_1},{v_3}}. Then (if I am interpreting the definition correctly) |{(u) \in E_g: u \in S}| = 2, so the empirical frequency equals 2/3. But it seems natural to calculate the empirical frequency as 3/3 = 1..? I think this definition is also needed for understanding the proof of Lemma 1 (please see "Minor Comments" below). Minor Comments: - Page 2, line 45: "...distinguishing class [gives] rise..." - Page 2, line 47: singletones -> singletons - Page 2, line 50: Missing article between "Thus," and "IPM". - Page 4, line 159: Should the vertex set \mathcal{V}_g be written without the subscript g? - Page 5, line 177: Missing conjunction between "g" and "a". - Page 5, line 178: I suggest adding brackets around k-n. - Page 5, line 187, definition of IPM_{\mathcal{D}(p_1,p_2): Superfluous closing square bracket in second expression and extra closing square bracket in third expression. - Page 5, line 190: \mathcal{V}^{{0,1}} -> {0,1}^\mathcal{V} ? - Page 6, definition 2: It might be helpful to summarise, in the main body of the paper if space permits or otherwise in the supplementary material, some basic properties of this new notion of VC dimension and how it relates to the usual notion of VC dimension. For example, if the graph VC dimension is an upper bound on the standard VC dimension (treating each concept as a set of k-hyperedges and defining shattering in the usual way), is there a Sauer-Shelah type upper bound on the size of the class in terms of the graph VC dimension? On a related note, on page 3, line 97, perhaps one could write precisely what it means for the new notion to be "strictly weaker" than the standard VC-dimension; does it mean "at least equal to"? I also suggest using an alternative term (perhaps "hypergraph VC dimension"?) since the term "graph dimension" seems to have already been used for a notion of VC dimension for multivalue functions (e.g. in the paper "Characterizations of learnability for classes of {0,1,...,n}-valued functions" by Ben-David, Cesabianchi, Haussler and Long)... - Page 7, line 232: class -> classes - Page 7, line 251: Missing space between "bounds" and "[1]". - Page 7, line 261 and Page 17, line 528: Missing period after "1/2". - Page 8, line 295: "...the VC [dimension] of a class..." - Page 10, line 369: [Preliminaries] - Page 10, line 372: characterizes -> characterize - Page 10, footnote 4: lable -> label, missing period at the end of the sentence. - Pages 11 and 12: In the statements of Theorem 2 and Lemma 1, the distribution over the vertices is denoted by P; in the inequality between lines 406 and 407, as well as in the proof between lines 408 and 422, the distribution is denoted by D. Do they refer to the same distribution (seems so)? - Page 11, line 417: "...let S_{i,+} [be] all the..." - Page 12, 5th line of equalities/inequalities: Extra vertical line just before the second "sup". Should the fifth line start with an inequality rather than an equality..? (If not, why?) - Page 13, third line in chain of equalities/inequalities just after "Bounding *": Under the second expectation symbol, should v_{1:k} be v_{1,k-1} ? - Page 13, line 429: "...uniform distribution over k-tuples from..." -> (k-1)-tuples? - Page 13, line 430: "...according to d and..." -> "...according to D and.." ? - Page 13, line 431: element -> elements - Page 13, lines 429 to 433: I was a bit confused at this part of the argument. It seems that S is a sample with possibly repeated vertices, so putting a uniform distribution over (k-1)-tuples of S where each (k-1)-tuple is assigned a probability of 1/m^{k-1} seems to mean that two k-tuples with equal corresponding elements but different sets of corresponding indices are considered to be different? E.g. if S = {v_1,v_2,v_3} is a sample such that v_1 = v_2 = v_3, and k-1 = 2, then (v_1), (v_2) and (v_3) are each assigned a probability of 1/3. If so, why is the process described as one "where we pick m elements according to D and then partition them to m-k+1 elements and to a sequence v_{1:k-1} of distinct [elements]?" In particular, why does v_{1:k-1} consist of distinct elements (i.e. none of its terms are equal?); is this an assumption, and if so, why is such an assumption needed? Why does this process give rise to a "partition" of S unless v_{1:k-1} is picked from S without replacement (is this what was meant)? - Page 14, line 438 and the inequality just after line 439: \mathcal{D}_{u_{1:k-1}} -> \mathcal{G}_{u_{1:k-1}} ? - Page 14, lines 438 and 440: Why is gVC(\mathcal{G}_v) exactly \rho for any choice of v (since, in the definition of gVC, we are taking the maximum projected VC dimension)? Should it be less than or equal to \rho? * Response to author feedback Thank you very much for the detailed feedback. I understand the paper a bit better now, and I think it deserves a higher score. I have therefore increased my overall score to 8. (I just recommend proofreading the paper for minor typos etc....)

Reviewer 3



Due to the shortness of time, I only comment upon the expressivity results. The expressivity results are interesting and novel. However, I found it rather hard to sort out the big'' picture of these results, and perhaps some additional general discussion would be helpful. It seems that in this part of the paper a kind of partial ordering between distinguishing classes is studied, where a class is larger than another if it can distinguish more pairs of distributions (classes can also be equivalent, and because of the parameters it is not clear how to formalize this precisely). Theorem 4, for example, shows the superiority of using k=2 over k=1. The graph class is a single graph (an infinite bipartite graph encoding all finite subsets of one half by vertices of the second half). It is shown that for any finite-VC class there are distributions distinguished by the graph but not the class. In terms of ordering the theorem says that there is a class for $k=2$ (consisting of that single graph) which is not smaller than or equal to any finite VC-class for $k=1$. The term incomparable''is used at a few places and it is not clear in what sense it is meant. Also, as the ultimate goal of the work seems to be to find possibly simple classes which can distinguish many pairs of distributions, it would be useful to comment on what are the implications of such incomparability results towards this goal.

[Author Response · NeurIPS 2019]

We thank the reviewers for the feedback and comments, in what follows we address specific comments made by the reviewers

**Reviewer 1**

**I do not completely understand (apart for some parts of the proofs) why refer to these functions as Graph-based.**
Boolean k-ary functions may be thought of as hyper-graphs. To some this point of view may be confusing and to some it may be helpful. Our original intuition was that certain constructions from graph theory will be helpful in the construction of some lower bounds (specifically in the expressiveness section) – hence we defined the model in terms of graphs. Since the submission we have made certain intermediate progress that seem to confirm our original intuition that the graph-based point of view is indeed helpful, from a technical perspective. But we agree that in the current paper this did not come into play, and we might tone down the graph-viewpoint in the final version.

**Reviewer 2**

**I was less certain about the significance of the result for finite domains; if there are natural and important examples of distinguishing classes with sublogarithmic sample complexity, it might be good to mention them somewhere in the paper** It is true that in many natural cases the VC dimension scales at least logarithmically in the domain size, nevertheless our point here was to quantitatively assess the previous result, note that we did not prove a lower bound, so apriori it might be that we can give better constructions that yield maybe even linear sample complexity.

**The definition of the empirical frequency of an edge (between lines 176 and 177 on page 5) seems a bit unusual if I am interpreting it correctly ...** The definition shouldn't be unusual and it will be clarified to avoid any possible confusion. We count the probability of an edge given $k$-iid vertices drawn from the empirical sample (with repetitions): $\frac{1}{m^k} \sum_{u_{1:k} \in S^k} g(u_{1:k})$. This is completely analogous to the standard empirical distribution for hypotheses classes.

In the example you write $v_1 = v_2$ but $v_1 \in E_g$ yet $v_2 \notin E_g$. this seems a bit confusing. How can $v_1 = v_2$ but one is in $E_g$ and the other not, can you please elaborate on the example so we may understand the confusion? Again, we will make sure to clarify this point.

**It might be helpful to summarise, ..., some basic properties of this new notion of VC dimension... ..., is there a Sauer-Shelah type upper bound on the size of the class in terms of the graph VC dimension?** We will be happy to elaborate on these in the main text. We have succinctly discussed the relation to the VC dimension (Here VC dimension refers to the VC dimension of a family of k-ary Boolean functions, considered as Boolean functions over the domain $X^k$). The VC dimension upper bounds the graph VC dimension (and that is what we mean by weaker: Small VC dimension entail small graph VC dimension). Regarding Sauer Shelah: It is kind of surprising but there is no Sauer Shelah Lemma for graph VC dimension, indeed this is noteworthy and we should discuss this in the main text.

**I would have also liked to know more about the impact of this work on the study of GANs** Discrimination plays a crucial role in the GANs setting, where a discriminator that observes data from true and synthetic distributions and needs to distinguish. Normally, the discriminator trains a neural net to distinguish the distributions – this is as done in the standard learning setting of classification. This work is the first to, theoretically, study discriminators of higher order types that observe multiple points (such empirical works were in fact considered, see related work).

**Reviewer 3**

**I found the paper difficult to follow and I am concerned by the lack of simulation studies and real application.**
A possible very real application is GANs, where discriminators are widely used during training. Since our results are mostly theoretical, studying the expressiveness of different models, simulations would not indicate the theoretical results we are searching for.

[Meta-Review · NeurIPS 2019]

Except for an outlying reviewer 3/4 reviewers agree that this a strong paper. A nice generalization od VC-dimension is given upper and lower bounds are proven. Recommended to be accepted.